Towards Measuring Resilience of Flood Prone Communities: A Conceptual Framework
V.O.Oladokun[1] and B.E.Montz[2]
[1]Department of Industrial and Production Engineering, University of Ibadan, Ibadan, Nigeria
[2]Department of Geography, Planning, & Environment, East Carolina University, Greenville, NC, USA

**Abstract**

Community resilience has become an important policy and research concept for understanding and
addressing the challenges associated with the interplay of climate change, urbanization, population
growth, land use, sustainability, vulnerability and increased frequency of extreme flooding.
Although measuring resilience has been identified as a fundamental step toward its understanding
and effective management, there is, however, lack of an operational measurement framework due
to the difficulty of systematically integrating socio-economic and techno-ecological factors. The
study examines the challenges, constraints and construct ramifications that have complicated the
development of an operational framework for measuring resilience of flood prone communities.
Among others, the study highlights the issues of    proliferation of definitions and conceptual
frameworks of resilience, challenges of data availability, data variability and data compatibility.
Adopting the National Academies' definition of resilience, a conceptual and mathematical model
was developed using the dimensions, quantities and relationships established by the definition. A
fuzzy logic equivalent of the model was implemented to generate resilience indices for three flood
prone communities in the US. The results indicate that the proposed framework offers a viable
approach for measuring community flood resilience even when there is a limitation on data
availability and compatibility.
Keywords: Hazard, Disaster, Flood, Resilience, Measurement, Fuzzy, Community

**1.0 Introduction**

Developing resilience of communities has become widely recognized as critical for disaster risk management due to the increased incidents of extreme weather events, such as flooding, which have disrupted economic activities, caused huge losses, displaced people and threatened the sustainability of communities across the world (Cai et al., 2018; Cutter 2018; Mallakpour and Villarini, 2015; Montz, 2009; Oladokun et al., 2017; Su, 2016a; Wing et al., 2018). Major international policy instruments such as the United Nations International Strategy for Disaster Reduction's (UNISDR) 2015 Strategic Framework and the 2005 Hyogo Framework have emphasized and adopted resilience principles in disaster risk management (Cai et al., 2018; Cutter et al., 2016). For instance, the interplay of extreme floods, population growth and rapid urbanization has increased flood hazard risks such that conventional flood risk management (FRM) measures of concrete structures, levees, flood walls and other defenses have become inadequate and unsustainable across various communities (Duy et al., 2018; Guo et al., 2018; Trogrlić et al., 2018; Wing et al., 2018). Resilience has gained a lot of attention, from both policy and research perspectives, involving using it to understand and address the challenges of land use, vulnerability and sustainability in the context of flooding (Cohen et al., 2016; Cohen et al., 2017; Folke, 2006; Parsons et al., 2016; Sharifi, 2016). Building community resilience has emerged as particularly relevant in dealing with flooding, which has become the most widespread and destructive of all natural hazards globally (Jha et al., 2012; Mallakpour and Villarini, 2015; Montz, 2009).

Consequently, there has been a shift from relying solely on large-scale flood defense and structural systems towards an approach that emphasizes the concept of community resilience as a strategic component of flood risk management (Hammond et al., 2015; Park et al., 2013). This shift is being reinforced by a consensus that since floods cannot be all together prevented; FRM must focus more on building the resilience of flood prone communities (Joseph et al., 2014; Oladokun et al., 2017; Schelfaut et al., 2011).

There is a consensus that the first and fundamental step toward understanding and operationalizing resilience for flood disaster and hazard management is to have an acceptable resilience measuring template (NRC, 2012). For instance, the ability to understand and objectively evaluate the impact of FRM programs, interventions and practices on community flood resilience is needed for making

political and business cases for proactive FRM investment from both public and private sectors.
Cutter (2018) suggested that an acceptable template is a basic foundation for monitoring baselines
and progress in building hazard resilience.
Furthermore, a measuring template will be useful as a decision support tool for the efficient
deployment of scarce FRM resources and also provides a basis for monitoring resilience changes
with respect to resource deployment.  For instance, Keating et al. (2017) explained that there is a
need for the continued development of theoretically sound, empirically verified, and applicable
frameworks and tools that help in understanding key components of resilience in order to better
target resilience-enhancing initiatives and evaluate the changes in resilience as a result of different
capacities, actions and hazards.
Therefore, the search for an acceptable framework and empirical model for measuring resilience
remains relevant and continues to attract attention (Cutter et al., 2016; Zou et al., 2018;   Cai et al.,
2018; Keating et al., 2017). Some existing measuring approaches, as identified in Cai et al., (2018),
include the Baseline Resilience Indicators for Communities (BRIC), the Resilience Inference
Measurement (RIM) framework, the National Oceanic and Atmospheric Administration (NOAA
2010) Coastal Resilience Index, the PEOPLES Resilience Framework, and the Communities
Advancing Resilience Toolkit (CART). There is also the '5C-4R' Zurich Alliance framework
combining the 'five capitals' of the UK's Department for International Development  DFID
sustainable livelihoods framework (Scoones, 1998) and the four properties of a resilient system
(Szoenyi, et al., 2016): the framework incorporates a technical risk grading standard (TRGS)
developed by Zurich risk experts  (Keating et al. 2017).
Despite the attention resilience has gained, the concept remains difficult to operationalize in the
context of community flood risk management due to, among other factors, the difficulty in
measuring resilience (Cutter, 2018; Fisher, 2015). Many experts and authors have noted  the
difficulty in integrating indicators of the natural and human systems as well as socio-environmental
factors into resilience by most of the existing frameworks (Cai et al., 2018; Cutter, 2018; Fuchs
and Thaler, 2018; Qiang and Lam, 2016).  Resilience, as a multifaceted and multidimensional
concept, has developed across multiple disciplines and applications such that resilience discourse
has attracted multidisciplinary interests from both research and policy perspectives.  While the
wide spectrum of multidisciplinary and practice interests characterizing resilience discourse has
increased its understanding and generated insights, it has also led to the emergence of multiple
variants of its definiton as well as the absence of consensus on the conceptual framework for its
measurement (Brown and Williams, 2015; Cohen et al., 2016; Cutter 2018). For instance,
resilience has been noted to have varied definitions depending on the hazard and disciplinary
contexts, with over 70 definitions identified by Fisher (2015).
The multiplicity of definitions has led to proliferation of conceptual models, frameworks and
interpretations (Costache, 2017), such that there is difficulty in transforming resilience
measurement from an abstract concept into an objective operational quantitative template.
According to Cutter (2018), the difficulties in harmonizing and operationalizing these definitions
have led to the emergence of a wide array of measurement approaches. Meanwhile, a pre-requisite
to having an operational model, in the context of resilience measurement, is the adoption or
convergence of definition by the resilience research and policy community. Such a definition
should meet the following criteria: i) emanates from or receives the formal endorsement of a
widely recognized institutional platform of stakeholders, ii) encompasses a wide spectrum of
existing resilience concepts, iii) has some degree of simplicity, and iv) enjoys high acceptance of
both the research and policy community. In a widely cited National Research Council report
(NRC, 2012), the US National Academy of Sciences defines resilience as the ability of a system
to prepare and plan for, absorb, recover from, and more successfully adapt to adverse events (Cai
et al., 2018; Cutter, 2018). Therefore, this study has adopted this definition as the basis for the
proposed framework for measuring the resilience of flood prone communities.
From a systems perspective, community-resilience is a non linear collection of socio-ecological,
socio-political, techno-ecological and socio-economic entities, each characterized by dynamic and
complex spatiotemporal interactions. Essentially, the concept of resilience involves the
interactions of several entities each defined by some social, economic, natural, technical and
environmental dimensions (Cai, et al., 2018; Norris et al., 2008). For instance, the community
component was succinctly described by Cai et al. (2018) as a coupled natural and human system
that manifests various sources of complexity such as nonlinearity, feedback, and uncertainty and
dynamic interactions.
Furthermore, coupled with the challenge of complexity and the dynamic nature of community-
resilience modeling is the challenge of data and computational analysis. It has been established

that information and data items characterizing community-resilience system are mostly imprecise, incomplete, vague, complex, fuzzy and subjective within the context of flood risk management (Kotze and Reyers, 2016,; Oladokun,et al., 2017). These characteristics present some operational and analytical challenges for any complex model based on traditional crisp mathematics and hard computational approaches   because of data availability, data variability and data compatibility. The resilience measuring problem with its interplay of definitional ambiguities, multi-dimensionality, and spatiotemporal dynamics invariably results in complex mathematical models. Such models, given the level of incompleteness, vagueness, and subjectivity that characterizes the human and socio-political aspects of resilience, offer little tractability with conventional hard computational tools and are difficult to operationalize. Hence, Oladokun et al. (2017) suggested that a resilience measuring model may be more amenable to a soft computing analytical technique such as fuzzy logic.

## 1.1 Aim and objectives

Based on the background presented above, this study is aimed at adopting a soft computing approach, a fuzzy logic computational model, for the proposed flood resilience measuring template. In particular, the objectives of  the study are  1) the development of a descriptive model that outlines our abstract interpretation of community resilience as a system, using insights from relevant literature, interactions with  experts  and observations of selected flood prone communities, 2) development of an equivalent  mathematical model of the resulting descriptive model using an appropriate tool to generate further insights, and 3) development of an equivalent fuzzy inference system suitable for  computational and  analytical purposes in the face of the aforementioned data  issues.  The next section briefly describes some relevant fuzzy logic concepts.

## 1.2 An Overview of Fuzzy Logic

Fuzzy set theory provides a mathematical tool for modeling uncertain, imprecise, vague and subjective data which represents a huge class of data encountered in most real-life situations (Adnan et al., 2015; Lincy and John, 2016). The fuzzy logic (FL) concept, introduced in 1965 by Lot A. Zadeh, is an extension of the classical set theory of crisp sets. FL, like humans, accommodates grey areas where some questions may not have a clear Yes or No answer or black and white categorization. According to Zadeh (1996), Fuzzy Logic = Computing with Words. FL mimics human reasoning and capability to summarize data and focus on decision-relevant

information in problems involving incomplete, vague, imprecise or subjective information. It is a
computational concept that allows for modeling of complex systems using a higher level of
abstraction originating from our knowledge and experience. It provides a very powerful tool for
dealing quickly and efficiently with imprecision and nonlinearity (Oladokun and Emmanuel,
2014). This capability to mine expert knowledge and use limited or fuzzy data makes fuzzy
inference systems (FIS) a suitable tool for resilience measurement modeling.
The concept of membership function (MF) is central to FIS. In traditional logic, an element $x$ is
either in or out of crisp set A; in other words, its degree of membership of the set is either zero or
one.   However, in fuzzy logic the element $x$ can be in a fuzzy set B 'partially' by using a MF
$\mu_B(x)$ which can return any real value between 0 and 1. This returned value is the degree of
membership representing the degree to which the element belongs to a fuzzy set. Therefore, in FL,
the truth of any statement becomes a matter of degree.
Thus for crisp set A  $\mu_A(x) = \begin{cases} 1 & if\ x\ \in A \\ 0 & otherwise \end{cases}$
On the other hand, for a fuzzy set, the MF may be represented as follows
$\mu_B(x) = \begin{cases} f(x) & if\ b_1 \leq x \leq b_2 \\ g(x) & if\ b_2 < x \leq b_3 \\ 0 & otherwise \end{cases}$
Actually, the crisp set is a special case fuzzy set whose MF returns only zero or one. There are
many functions that are used as MFs. Some widely used MFs, Generalized bell shaped, Gaussian
curves, Polynomial curves, Trapezoidal, Triangular and Sigmoid MFs (Oladokun and Emmanuel,
2014; Adnan et al., 2015). The Mamdani FIS approach (Mamdani and Assilian, 1975), adopted
for this study, is made up of a fuzzy inference engine characterized by the use of carefully selected
MFs and a fuzzy rule base. The rule base is a set of 'IF THEN' statements that capture experts'
knowledge of the logic governing the problem.  The fuzzy inference system will provide a template
for experts and other stakeholders to translate their perceptions of the  problem and map their
linguistics rating of these variables   into a resilience index based on the fuzzy relationships we
define.

**2.0 Resilience Measuring:  A Conceptual Framework**

**2.1 Descriptive model**

The design objective is to have a conceptual framework and its associated mathematical model with sufficient tractability by minimizing the number of model elements and adopting the barest minimum relationships while maintaining a reasonable level of validity. Therefore, as the theoretical basis for the proposed conceptual model, as mentioned earlier, we are adopting the resilience definition put forward by the US National Academies (NRC 2012). Conceptually this definition implies that a community's resilience is a quantity that reflects capacities such as: 1) the community's coping capacities, in terms of a threshold of hazard it can absorb (Hazard Absorption Capacity H), 2) its accessible resources (Resource Availability G), and 3) its resource utilization efficiency determined by factors like its preparedness and its governance processes (Resource Utilization Processes $\theta$). These capacities interact to define its ability to prepare for, absorb, recover from, and more successfully adapt to adverse flooding events. We attempt to conceptualize this understanding as shown in Figure 1.

Each of the dimensions in Figure 1 is influenced by a number of technical, social, ecological, economic, and political factors following work that has been reported in the literature which sheds light on these factors and how they influence the dimensions (see Cohen et al., 2016; Lee et al., 2013; Rose, 2017). For example, hazard absorbing capacity H is determined by a number of techno-ecological factors such as adequacy, sophistication and use of infrastructure and technology as well as redundant capacities. It is also determined by socio-ecological and socioeconomic factors that influence both individual and institutional coping capacities. Resource availability is determined by things like community capital, political influence, and economic activities as well as ecological resources accessible to drive the quality and timeliness of recovery. Resource utilization processes are determined by the quality of governance and institutions such as judiciary, police, media, and public service. These processes influence policy formulation and implementation, the ease of doing business and the efficiency of use of resources. A detailed structured and operational rendition of the foregoing is presented in sections 2.2 and 3.3.

Figure 1 here

Furthermore, in the context of FRM, the framework of Figure 1 recognizes that resilience enhances
recovery or that recovery is an outcome of resilience whereby when a community, as a coupled
system, becomes more resilient its capacity to experience post disaster recovery increases. In other
words, recovery, in terms of time taken to attain post disaster recovery and the degree of recovery
attained, is influenced by its resilience. Invariably the conceptual framework implicitly suggests
that recovery (recovery speed and recovery quality) can surrogate resilience. This is reasonable
because post disaster recovery is driven by resilience factors such as preparedness, and coping
capacity, among others. This understanding is supported by the DROP disaster resilience model
of place (DROP) as illustrated in Cutter et al. (2008), reproduced in Ffigure 2.
Figure 2 here
**2.2 Mathematical model**
The next stage is to transform the conceptual framework of Figure 1 into an operational
mathematical model. This is accomplished by defining a geometric model of the framework as
shown in Figure 3. This model is then used to derive appropriate mathematical relationships for
resilience measurement and provide some insights.
**2.2.1 Notations, definitions and terms**
We adopt the following notations, definitions and terms   to explain the components of Figure 3 in
the context of flood hazard.
i.    Hazard Absorbing Capacity ($H$): ($H=h$: $0\leq h \leq 1.0$). The resilience of a community
depends on the level of the flood hazard the community systems can absorb before
totally collapsing or undergoing irreversible disintegration. $H=1$ is the highest
absorbing capacity whereby the community can absorb and survive the damages and
disturbance (both structural and non structural) of the most severe category of flooding
conceivable. This captures various resilience factors such as coping capacity,
redundancy, preparedness, sense of place attachment and other capacities as explained
in Table 1.
ii.   Resource Availability ($G$).  This is the quantum of resources available to plan and
pursue recovery as well as achieve recovery quality level $Q$ (including adaptive
recovery). Note that $G=g$ ($0\leq g \leq 1.0$) captures both economic and community capital.
It is the measure of resources the community is able to attract as a result of its overall

| 233 | | economic and political influence, its natural assets, and human capital assets (see Table |
| 234 | | 1 for further details). |
| 235 | iii. | Resource Utilization Processes ($\theta$): With $0 \leq \theta \leq \Pi/2$, we define $\rho$ ($\rho = \mathrm{Sin}\ \theta$) as system |
| 236 | | efficiency. This is a resilience component that affects recovery and revolves around |
| 237 | | factors such as preparedness, community governance, institutional systems and |
| 238 | | processes. It determines the efficiency and effectiveness of the use of resources to |
| 239 | | achieve recovery and establish adaptive capacity. In other words, how *well* resources |
| 240 | | are used is as important as how *much* of a set of resources is used in building resilience. |
| 241 | | It measures the probity, level of accountability, level of waste, corruption, red-tapism, |
| 242 | | and bureaucracies within the system. A community with strong institutions such as a |
| 243 | | functioning judiciary and an efficient civil service, for instance, will tend to return high |
| 244 | | $\rho$. So an ideal or utopian community will have its $G$ deployed at $\theta = \Pi/2$, such that $\rho =$ |
| 245 | | $\mathrm{Sin}\ (\theta) = \mathrm{Sin}\ (\Pi/2) = 1$. |
| 246 | iv. | Recovery Quality Level ($Q$). This represents the outcome of post hazard conditions in |
| 247 | | terms of restoration quality and socio-ecological functionality, among others. |

| 248 | The following definitions apply with reference to Figure 3 |

| 249 | v. | $a_i$ : Resilience reservoir of a real system i is defined as the area of trapezium ABFE' |
| 250 | | determined by the hazard absorbing capacity, at $H = h$, of the system, the available |
| 251 | | quantum of resources ($G = g$), the quality of governance processes and resource |
| 252 | | utilization systems ($\mathrm{Sin}\ \theta$) and the achievable recovery quality ($Q = q$) |
| 253 | vi. | $a_u$ : The resilience reservoir of an utopian (ideal) system is defined as the area of square |
| 254 | | ACDE. This occurs at ideal FRM conditions: that is, a community system with |
| 255 | | adequate resources, perfect governance and processes with zero waste of resources and |
| 256 | | infinite hazard coping threshold when h= AE (or at maximum absorbing capacity), |
| 257 | | g=ED (maximum resource adequacy) and $\theta = \Pi/2$ (perfect or utopian system with |
| 258 | | 100% efficiency or $\mathrm{Sin}\ \theta = 1.0$). The utopian system can achieve a perfect recovery |
| 259 | | index Q= q= 1.0 or Q=AC |

| 260 | Extensive review of the literature was carried out to provide an informed basis for mapping |
| 261 | FRM factors and inputs to the dimensions of resilience. This is summarized as shown in Table |

1. Theoretically, the values of the dimensions H, G, θ can be estimated from adequate data on
these input factors and appropriate functions.
**Table 1 Resilience Dimensions Input Factors**

| Resilience Dimensions | Resilience input factors |
|---|---|
| 1. Hazard Absorbing capacity H | 1. Level of infrastructure in terms of sophistication and adequacy. Effectiveness of FRM measures such as flood and shoreline defenses, forecast and warning system,<br>2. Redundant capacities. Evidence of alternatives in critical utilities, evacuation routes, communication and energy infrastructures, hospitals, police posts, supermarkets.<br>3. Evidence of redundant housing capacity.<br>4. Ecological defenses and buffer. Evidence of complementary use of nature to improve threshold, e.g. using landscaping and topography, natural drainage and canals, vegetation cover, rain/storm water harvesting, permeable pavements, etc.<br>5. Residents coping capacity. Evidence of large portion of populace with previous flood experience, awareness, cohesion and place attachment<br>6. Evidence of stable or growing population in spite of past events.<br>7. Educational and literary level of populace<br>8. Evidence of social and communal clusters to enhance coping through support, meaning, avoidance etc., e.g. church, local sport team, ethnic clusters.<br>9. Presence of critical and strategic institutions of national importance, e.g. university, military base, major ports, etc.<br>10. Evidence of technology driven information dissemination, e.g. social media, sms (Ashraf and Routray, 2013; Cohen et al., 2017; Esteban et al., 2013; Ibanez et al., 2004; Lee et al., 2013; Mavhura et al., 2013) |
| 2. Resource Availability G | 1. Evidence of budgetary provision for, or commitment to, flood risk management.<br>2. Evidence of thriving economic activities in the community, e.g. size of local GDP<br>3. Evidence of economic strength of residents, e.g. per capita income, income level, housing value, savings, cooperative societies, etc.<br>4. Evidence of political, institutional and economic influence that can attract grants and funds from national or regional sources, e.g. population<br>5. Evidence of adoption of flood insurance plans.<br>6. Availability of land for relocation development beyond or outside the flood plains.<br>7. Evidence of community capital and community natural assets accessible for reconstruction, e.g. forest resources, granite and quarry deposits.<br>8. Economic status of the 'parent' entity, e.g. the state's or country's GDP (Filion and Sands, 2016; Rose, 2017; Swalheim and Dodman, 2008; Thomas and Mora, 2014) |
| 3. Community Processes and Resource Utilization θ | 1. Evidence of good governance<br>2. Level of ease of doing business<br>3. Evidence of strong institutions such as judiciary, police, media, and public service<br>4. Evidence of culture of law and order.<br>5. Ranking of internationally recognized bodies like Transparency International, World Bank, UN, CIA, etc. on the above (Begg et al., 2015; Brown and Williams, 2015; Cohen et al., 2016; Rose, 2017; Tompkins et al., 2004) |


**Table 1 here**
Figure 3 here

## 2.2.2 Resilience modeling

The utopian resilience reservoir is the benchmark for evaluating resilience such that actual
resilience $R_i$ can be defined as the ratio of $a_i$ to $a_u$ as indicated in equation 1.
$R_i = \frac{a_i}{a_u}$       (1)
Using the insights from Figure 1, we attempt to develop the mathematical model implied in
equation 1 (note R is dimensionless since both $a_i$ and $a_u$ are areas).
$a_i = \frac{1}{2}\{AE' + BF\}AB$       (2)
$a_u = AE \times ED$
$a_u = H \cdot G$       (3)
Note: $AE' \equiv h$       (4)
$BF = AE' - F'E' = h - gCos\theta$       (5)
$AB = F'F = gSin\theta$       (6)
Putting 4, 5, 6 into 2
$\Rightarrow a_i = \frac{1}{2}\{h + (h - gCos\theta)\}gSin\theta$
$a_i = hgSin\theta - \frac{1}{2}g^2Sin\theta Cos\theta$
$a_i = hgSin\theta - \frac{1}{2}g^2Sin\theta \pm \sqrt{1 - Sin^2\theta}$
Recall we define 'Efficiency of resource utilization system' as $\rho = Sin\theta$
$\therefore a_i = hg\rho - \frac{1}{2}g^2\rho\sqrt{(1 - \rho^2)}$       (7)
Putting 3 and 7 into 1
$R_i = \dfrac{hg\rho - \frac{1}{2}g^2\rho\sqrt{(1 - \rho^2)}}{HG} -$       (8)
Without loss of generality, h and g are treated as indices such that
$0 \leq h \leq 1 \quad and\ 0 \leq g \leq 1$
Then $H=G=1$ in equation 8 which implies
$R_i = hg\rho - \frac{1}{2}g^2\rho\sqrt{(1-\rho^2)}$          (9)
Equation 9 is a valid expression for resilience.
That is, $R_i = f(h,g,\rho)$,
Where h, g and h are as explained in section 2.2.1 and their values ─are decided by experts and/or
stakeholders, varying depending upon the location and scale of application of the model.
**2.2.3 Some insights from model using some extreme values**

This section discusses some example cases of the model (equation 9) output using selected
hypothetical extreme parameters' values  to generate further insights into model structure (with
reference to Figure 1). The 'extreme' scenarios analysis is used to demonstrate how each of the
three 3 dimensions impacts R.
**Case 1: As  $\rho \rightarrow 0$     $R \rightarrow 0$**
In fact, R= 0 when $\rho = 0$. This may be interpreted as the case when the resource utilization
processes have zero efficiency (see Figure 4) or a collapsed governance system such as when a
flood disaster occurs in a community ravaged by civil war with breakdown of law and order. In
such situations, community resilience is nil as all resources put into recovery will be 'wasted,'
irrespective of the level of coping or infrastructure previously in place.

Figure 4 here

**Case 2: As  $\rho \rightarrow 1$    $R \rightarrow hg$**
This implies that θ=Π/2 or Sinθ=1 which depicts an ideal situation when the communal processes,
FRM resource administration, and utilization systems are highly efficient and near perfect.  Under
this scenario, the resources g and community's coping capacities contribute maximally to
resilience (see Figure 5).

Figure 5 here
**Case 3: $g \to 0$     $R_i \to 0$**     Resilience disappears when resources dry up.

**Case 4: h= 1**   Resilience is determined by resource availability and utilization

**Case 5:  As  $h \to 0$     $R \to 0^{-}$**
From Figure 6, resilience approaches zero from negative reservoir quadrant when h=0 (i.e. coping
and absorbing capacities disappear or collapse) and $\rho < 1$ (efficiencies of resource use,
preparedness, and governance systems fall below 1). The 'Negative' resilience reservoir quadrant
characterizes vulnerable communities. Note that vulnerability is sometimes seen as the flip side of
resilience (Folke et al., 2002) or a complementary community-hazard management concept
(Cutter, 2018; Fekete and Montz, 2018; Shah et al., 2018). Hence from figure 6 as the
absorbing/coping capacity h approaches zero, a community enters vulnerability mode because
more resilience area lies below the positive plane. In other words, equation 9 suggests that a
community without coping or built in absorbing capacities is vulnerable, especially if its
governance structure is poor (i.e. Sinθ $\to$ 0).

Figure 6 here

**3.0 Resilience fuzzy inference system (R-FIS):  Computer model**
While the resulting model of equation 9 provides useful insights, its application however is
premised on the availability of clear information on input factors and adequate data for estimating
model parameters, That is, complete data as described in section 2.2 and Table 1, for estimating
dimensions H, G and θ.  However, there are issues of data availability and data compatibility
(Parsons et al., 2016) which make it inefficient to do crisp estimation of these parameters.
Therefore, to operationalize the proposed framework, a (FIS) equivalent has been developed.
A computer model of the proposed R-FIS (Figure 7) was designed in the Matlab fuzzy logic
development environment. The environment was adopted because it supports easy to use graphical
user interface (GUI) tools and has multiple MFs for implementing a FIS. A process consisting of
systematic review of the literature, interactions with experts, meetings with community leaders,
interviews of other stakeholders and field observations (described in more detail in Section 4.1)

was -used to gain insights for specifying the R-FIS's design and inference engine's elements (Table 2) as well as determine appropriate IF THEN statements for the rule base (Table 3). With three input linguistic variables, each with three term sets (or possible values), there can be up to 27 explicit input variable combinations, or 27 explicit fuzzy rules combinations. Table 3 is a sample extract from the 27 'IF THEN' statements of the rule base.

**Commented [DVO2]:** .. The response to (reviewer 2, comment to section 7) has been provided in the section 4.1 of the revised manuscript. We believe that is the most appropriate place to explain our methods in detail, as this section addresses the model itself.

Figure 7 here

Table 2 here

**Table 2 Fuzzy Inference Linguistic Variables Term set and Membership Functions**

| Linguistic Variables | Term sets | Membership function |
|---|---|---|
| Hazard Absorbing Capacity H **Input 1** | Low | PiMfunction |
| | High | GbellMf |
| | Very High | SMfunction |
| Resource Availability G. **Input 2** | Very Low | ZMfunction |
| | Low | GaussianMfunction |
| | High | SigMfunction |
| Resource Utilization Processes θ. **Input 3** | Poor | PiMfunction |
| | Good | GaussianMfunction |
| | Excellent | PiMfunction |
| Resilience R. **Output** | Very Low | Zmfunction |
| | Low | Gauss2Mfunction |
| | Moderate | GbellMfunction |
| | High | PiMfunction |
| | Very High | PiMfunction |

**Table 3: Sample rules of the R-FIS 27 Rule Base***

| Rules premise | Rules Consequence | Weight |
|---|---|---|
| If (**H** is Low) & (**G** is Very Low ) & (θ is Poor) THEN | (Resilience is very low) | 1 |
| If (**H** is Low) & (**G** is Low) & (θ is Excellent ) THEN | (Resilience is Low) | 0.8 |
| If (**H** is Low) & (**G** is High) & (θ is Excellent) THEN | (Resilience is Moderate) | 0.8 |
| If (**H** is High) & (**G** is High) & (θ is Excellent) THEN | (Resilience is Moderate) | 1 |
| If (**H** is Very High) & (**G** is Very Low) & (θ is Good) THEN | (Resilience is High) | 0.7 |
| If (**H** is Very High) & (**G** is High) & (θ is Good) THEN | (Resilience is High) | 1 |
| If (**H** is Very High) & (**G** is High) & (θ is Excellent ) THEN | (Resilience is Very High) | 1 |

*Rules and weights to be determined by experts and/or stakeholders


Figure 8 shows the 3D surface plot resulting from an infinite combination of input factors. The
shape of the resilience surface is determined by the rules (Table 3) and the selected membership
functions (Table 2) used to express the term sets. This shape can be varied by modifying the
membership functions, the term sets, the rules and their weights to reflect new realities and
understandings about the resilience systems. This gives flexibility to simulate various
combinations of parameters in order to arrive at an optimum design.


Figure 8 here

**3.2. Model expert scoring framework**

Although information and explanations in Table 1, in principle, give a general guide for evaluating
and quantifying these dimensional inputs of the resilience model, there is still the need for an easy
to use operational template for capturing experts' input into the FIS in relatively standardized
fashion. Table 4 is an example of such an input template designed for this study. A typical
application procedure is described, in section 4.1, with the case study communities.


**Table 4 Linguistic Variables Input Template**

| Linguistic Variables Dimension | Tick the grey box next to your linguistic rating | | Tick the grey box that best reflects your score of your linguistic rating | | | | | |
|---|---|---|---|---|---|---|---|---|
| Hazard Absorbing Capacity (H) | Low | | 1 | | 2 | | 3 | |
| | Moderate | | 4 | | 5 | | 6 | |
| | High | | 7 | | 8 | | | |
| | Very High | | 9 | | 10 | | | |
| | | | | | | | | |
| Resource Availability (G) | Low | | 1 | | 2 | | 3 | |
| | Moderate | | 4 | | 5 | | 6 | |
| | High | | 7 | | 8 | | | |

| | | | | | | |
|---|---|---|---|---|---|---|
| | Very High | | 9 | | 10 | |
| | | | | | | |
| Resource Utilization Processes (0) | Poor | | 1 | | 2 | 3 |
| | Good | | 4 | | 5 | 6 |
| | Very Good | | 7 | | 8 | |
| | Excellent | | 9 | | 10 | |
| Location/city | | | | | | |
| Date of assessment | | | | | | |
| Assessors' name | | | | | | |

*Table 1 can be attached to this scoring template as a guide
**4.0 Model Application: Study location**
The following describes the application of the model using three flood prone communities in the
United State (U.S.). Following decades of experience in dealing with hazards and disasters, cities
and institutions in the U.S. offer considerable information and insights in community resilience
systems management (Su, 2016b). Two coastal states of North Carolina and Virginia are home to
many flood prone communities of various sizes with diverse socio-economic and techno-
ecological characteristics that readily lend themselves to a study of resilience. Both states have
adopted a number of FRM programs, policies, and strategies for building flood resilience across
many rural and urban communities– (North Carolina Floodplain Mapping Program, 2019;
Mogollón et al., 2016). Specifically, Norfolk, VA a coastal city in Virginia with a massive naval
base, Greenville, NC, a large university town, and Windsor, NC a small riverine rural town were
selected (Figure 9).   Table 5 summarizes some vital socio- economic features of these
communities.
Figure 9 here

Norfolk, located on the Chesapeake Bay and near several rivers, experiences precipitation
flooding, when the intensity of rainfall exceeds stormwater drainage capacity, storm flooding from
hurricanes and nor'easters, and tidal flooding due to its elevation and coastal location. Greenville,
with relatively flat topography is located on the Tar River and is traversed by a number of small
streams (Pitt County Development Commission, 2019). Besides riverine flooding, the relatively

flat topography of its coastal plain location leads to flooding from intense or long-lasting rain events such that the stormwater system is incapable of handling the overland flow. Located on the meandering Cashie River in eastern North Carolina, Windsor has experienced four major floods since 1999, all from tropical storms. Thus, not only are the communities different demographically, but they have rather different flood regimes and histories, with Windsor and Greenville experiencing riverine flooding, though with very different patterns of damage, and Norfolk experiencing a combination of coastal and riverine flooding.

**Table 5 Study Locations: Demographic and Topographic Summary**

| | Windsor NC | Greenville NC | Norfolk VA |
|---|---|---|---|
| Location type | Small town | City | Large city |
| Types flood | River/storm/ rain | River /storm/ Rain | Coastal /river rain/storm |
| Total Population | 3,630 | 84,554 | 242,803 |
| %Male | 59.3 | 45.8 | 51.8 |
| %Female | 40.7 | 54.2 | 48.2 |
| Median income * | 29,063 | 34,435 | 44,480 |
| Poverty rate * | 27.8 | 32.5 | 21 |
| Median Age | 38.6 | 26.0 | 29.7 |
| %Under 14 | 12.4 | 15.9 | 17.7 |
| %75 above | 8.7 | 4.3 | 4.6 |
| US Citizenship * | 97.9 | 96.8 | 96.6 |
| Non English speaking * | 5.83 | 6.74 | 10.3 |
| No of Households | 1088 | 36071 | 85485 |
| %Family household | 61.2 | 46.3 | 58.7 |
| Average household size | 2.29 | 2.18 | 2.43 |
| %Household with individuals above 65 | 34.1 | 14 | 20.3 |
| No of Housing units | 1193 | 40564 | 95018 |
| % of housing units occupied | 91.2 | 88.9 | 91.0 |
| Mean property Value ($)* | 93800 | 147100 | 193400 |
| ** Elevation (feet) | 25 | 56 | 30 |

*Source http:// census.gov

** United States Geological Survey Topographic Maps

Table 5 here

**4.1 Model application: data gathering and results**

For the purpose of illustration, input scores were developed using the template shown in Table 4 along with the guidelines in Table 1 and the communities' information, summarized in Table 5. The sample input data were generated based on the outcome of field studies and reflective

interactions with experts and stakeholders familiar with the study locations; these stakeholders
include academics, government officials and community leaders. In particular the sample scoring
was based on the insights derived from our understanding of their opinions, as well as demographic
and socio-economic information extracted from various historical and government records,
including the US census (Pitt County Development Commission, 2019; North Carolina Floodplain
Mapping Program, 2019; Mogollón et al., 2016). For instance, during a 2018 workshop by the
North Carolina Chapter of the American Planning Association held at Windsor, NC, the authors
had the opportunity to interact with and mine the knowledge of academics, students, city managers,
community leaders, relevant officials from emergency agencies, and curators of landmark centers,
among others. The authors also took tours of Norfolk, VA and Greenville, NC, under the guidance
of academics, GIS and FRM experts from the cities' universities.  These interactions and the
associated field studies provided insights for generating the sample scoring: the studies involved
interviews and qualitative characterizationassessment from site observations of community flood
control projects and individual property FRM retrofit systems. As an example, the perceptions of
resident planning experts and other stakeholders on how some ongoing flood risk management
interventions would have impacted the capacity of the community to cope with varying flood
levels was useful in classifying Hazard Absorbing Capacity, as was the extent and type of flood
control and retrofit projects.
Table 6 shows the results. Norfolk and Greenville both have relatively high hazard absorbing
capacities, with Norfolk rated as slightly lower owing to problems associated with the disruption
that regularly occurs from overland flooding combined with tidal flooding. Windsor's is lower
than Norfolk and Greenville but still moderate because of how the community has adapted to its
flood risk. Not surprisingly, Norfolk has the highest resource availability and Windsor the lowest
based on their size and relative wealth. At the same time, for the illustrative purposes here, size
and diversity of the communities are seen to be inversely related to resource utilization processes.
The model output, Resilience Index R, indicates that, based on the input values, Grenville's
resilience is slightly greater than Norfolk's while, not surprisingly, Windsor lags rather far behind.


Table 6 Input Scoring and R-FIS Resilience Index Output

| Experts Scoring Community | Model Input | | | | | | Model Output |
|---|---|---|---|---|---|---|---|
| | Hazard Absorbing Capacity (H) | | Resource Availability (G) | | Resource Utilization Processes (θ) | | Resilience Index R |
| | Linguistic Score | Score | Linguistic Score | Score | Linguistic Score | Score | |
| Norfolk, VA | High | 7.0 | High | 8.0 | Good | 6.0 | 0.836 |
| Greenville, NC | High | 8.0 | Moderate | 6.0 | Very Good | 8.0 | 0.9 |
| Windsor, NC | Moderate | 4.0 | Low | 2.0 | Very Good | 8.0 | 0.477 |

Table 6 here

The input to output mapping implemented in Matlab fuzzy toolbox allows for infinite combinations of input factors either by sliding or inputting the respective input variable axis on the fuzzy rule interface. Figure 10 is a snapshot of the input combinations for Greenville, using the scores from Table 6. The vertical bar (red line on each) can be moved to indicate how resilience changes with a change in one or another (or all) of the three variables. The yellow shapes indicate the rules (see the subset in Table 2) that contribute to each variable's score. All of the output, in both Table 6 and Figure 8, is based on expert insights and understandings and thus provides a dynamic template to measure resilience under different conditions. The proposed framework accommodates the understanding that community resilience should be treated as a multifaceted and multidimensional construct that can only be achieved by focusing on all aspects of a community system. While the fuzzy implementation of the  framework can be used both as a resilience index tool and  a resilience classification scheme, it is however, like many existing resilience measuring models, still dependent on the subjective opinions of experts and other stakeholders.

Figure 10 here

**5.0 Discussion and Conclusions**

Many previous studies have identified  the multiplicity of definitions (Costache, 2017; Fisher, 2015; Oladokun et al., 2017), as one of the major difficulties in transforming resilience

measurement from an abstract concept into an objective operational framework (Costache, 2017;
Fisher, 2015; Oladokun et al., 2017). This study proposes three criteria for adopting a suitable
definitional basis for a framework conceptualization. These criteria which address issues such as
the need to achieve model simplicity and accommodate the multidimensional nature of resilience
(Brown and Williams, 2015; Cohen et al., 2016; Cutter 2018), were used to recommend the
National Academies' definition of resilience (NRC, 2012) as a robust and viable basis for
developing a measurement model.
Similarly, many scholars have highlighted dealing with the complexity involved in the integration
of indicators of natural and human systems into a community resilience model (Cai et al., 2018;
Cutter, 2018; Fuchs and Thaler, 2018; Qiang and Lam, 2016) as a key to transforming resilience
measurement from an abstract concept into an objective operational framework. SpecificallyTo
that end, we adopt a 3three-component system to define the definition- conceptual model transition
in a way that reflects key relationships; among technical, social, ecological, economic, and political
factors; that have been reported in literature (Cohen et al., 2016; Lee et al., 2013; Rose, 2017) as
key to the multidimensional treatment of resilience.
Transforming the conceptual model into a quantitative template requires some sound theoretically
basis, a condition noted in Keating et al., (2017) as a prequisite for developing an acepatable
framework.  Hence this study recognizes that such a framework must show clear logical
relationships among the various indicators and dimensions of resilience and provide logical
linkages between their abstraction and empirical requirements. The geometric based mathematical
modeling approach we have adopted shows these relationships and provides the linkage between
conceptual model and operational requirements.  Based on this, mathematical functions were
developed to establish logical relationships among key socio-technical parameters and quantities
that characterize the community resilience system, thus infusing a theoretical basis into the
framework. To enhance the integration of both technical and non-technical communal resiliency
factors and reduce model complexity, the conceptual framework was defined using a minimum
number of integrated components and interactions. This approach allows the adoption of a soft
computing tool for model analysis.  While the study developed a template for data collection and
illustrated its application, the template still relies on subjective opinions of experts which may be
seen as a drawback of the model. Hence further research is suggested to explore the automation

and standardization of the R-FIS input process by integrating with web based socio-economic and ecological rankings or indices of communities. Yet, from computational and operational perspectives, the adoption of a fuzzy inference system as an analytical tool is presented as a viable approach for harnessing the opinions and experiences of experts and residents.

In conclusion, ~~This study~~this study ~~which~~ is centered on the need for an acceptable template to measure flood resilience. ~~As such, it~~ examines the challenges, conceptual constraints and construct ramifications that have complicated the development of an operational framework for measuring the resilience of communities prone to flood hazard.

Although the proliferation of conceptual models and frameworks for understanding resilience has indeed posed some challenges for development of an acceptable scenario-based measurement framework, there has been evidence of rich multidisciplinary insights resulting from the continuously evolving collaborative platforms for driving resilience research, policy and discourse. Non-linearity, multiple feedbacks and other sources of complexity constitute major challenges to achieving operational practicality and model tractability while maintaining reasonable validity. There has also been the challenge of compatibility between the natural and human variables due to the well recognized complexity inherent in community resilience. ~~The study recommends and adopts the National Academies' definition of resilience (NRC, 2012) as a robust and viable basis for developing a measurement model. Based on this, mathematical functions were developed to establish logical relationships among key socio-technical parameters and quantities that characterize the community resilience system, thus infusing a theoretical basis into the framework. To enhance the integration of both technical and non-technical communal resiliency factors and reduce model complexity, the conceptual framework was defined using a minimum number of integrated components and interactions. This approach allows the adoption of a soft computing tool for model analysis.~~

In terms of insights, the ~~resulting~~ models from this study provide some explanations into the relationships existing among resilience factors and dimensions. For instance, the importance of good community governance, processes and resource utilization systems becomes obvious in the various scenario analyses. Furthermore, the model was able to document the relative impact of variables that contribute to or detract from resilience. Although only sample values were used, the

model application was able to illustrate the relative impacts that varying levels of institutional strength and resource availability, for example, have on progress toward resilience at a place.

~~While the study developed a template for data collection and illustrated its application, the template still relies on subjective opinions of experts which may be seen as a drawback of the model. Hence further research is suggested to explore the automation and standardization of the R-FIS input process by integrating with web-based socio-economic and ecological rankings or indices of communities. Yet, from computational and operational perspectives, the adoption of a fuzzy inference system as an analytical tool is presented as a viable approach for harnessing the opinions and experiences of experts and residents.~~ Hence, ~~T~~the R-FIS provides a pathway for dealing with challenges of data issues such as missing data, spatiotemporal variations, and the use of subjective information because the critical input variables are locally and/or contextually defined. Thus, the proposed framework offers a viable approach for measuring flood resilience even when there are limitations of data availability and compatibility.

**Acknowledgements**

*This work is part of a research carried out under the Fulbright African Research Scholar Program Award (2017/18) funded by the United States Government.*

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

**Table 1 .** Resilience dimensions and descriptions of input factors influencing their states

| Resilience Dimensions | Resilience input factors |
|---|---|
| **1.** Hazard Absorbing capacity **H** | 1. Level of infrastructure in terms of sophistication and adequacy. Effectiveness of FRM measures such as flood and shoreline defenses, forecast and warning system, <br> 2. Redundant capacities. Evidence of alternatives in critical utilities, evacuation routes, communication and energy infrastructures, hospitals, police posts, supermarkets. <br> 3. Evidence of redundant housing capacity. <br> 4. Ecological defenses and buffer. Evidence of complementary use of nature to improve threshold, e.g. using landscaping and topography, natural drainage and canals, vegetation cover, rain/storm water harvesting, permeable pavements, etc. <br> 5. Residents coping capacity. Evidence of large portion of populace with previous flood experience, awareness, cohesion and place attachment <br> 6. Evidence of stable or growing population in spite of past events. <br> 7. Educational and literary level of populace <br> 8. Evidence of social and communal clusters to enhance coping through support, meaning, avoidance etc., e.g. church, local sport team, ethnic clusters. <br> 9. Presence of critical and strategic institutions of national importance, e.g. university, military base, major ports, etc. <br> 10. Evidence of technology driven information dissemination, e.g. social media, sms (Ashraf and Routray, 2013; Cohen et al., 2017; Esteban et al., 2013; Ibanez et al., 2004; Lee et al., 2013; Mavhura et al., 2013) |
| **2.** Resource Availability **G** | 1. Evidence of budgetary provision for, or commitment to, flood risk management. <br> 2. Evidence of thriving economic activities in the community, e.g. size of local GDP <br> 3. Evidence of economic strength of residents, e.g. per capita income, income level, housing value, savings, cooperative societies, etc. <br> 4. Evidence of political, institutional and economic influence that can attract grants and funds from national or regional sources, e.g. population <br> 5. Evidence of adoption of flood insurance plans. <br> 6. Availability of land for relocation development beyond or outside the flood plains. <br> 7. Evidence of community capital and community natural assets accessible for reconstruction, e.g. forest resources, granite and quarry deposits. <br> 8. Economic status of the 'parent' entity, e.g. the state's or country's GDP (Filion and Sands, 2016; Rose, 2017; Swalheim and Dodman, 2008; Thomas and Mora, 2014) |
| **3.** Community Processes and Resource Utilization **θ** | 1. Evidence of good governance <br> 2. Level of ease of doing business <br> 3. Evidence of strong institutions such as judiciary, police, media, and public service <br> 4. Evidence of culture of law and order. <br> 5. Ranking of internationally recognized bodies like Transparency International, World Bank, UN, CIA, etc. on the above (Begg et al., 2015; Brown and Williams, 2015; Cohen et al., 2016; Rose, 2017; Tompkins et al., 2004) |

**Table 2.** Fuzzy inference linguistic variables term set and membership functions (Adnan et al., 2015; Oladokun and Emmanuel, 2014)

| Linguistic Variables | Term sets | Membership function |
|---|---|---|
| Hazard Absorbing | Low | PiMfunction |
| Capacity H | High | GbellMf |
| **Input 1** | Very High | SMfunction |
| Resource | Very Low | ZMfunction |
| Availability G. | Low | GaussianMfunction |
| **Input 2** | High | SigMfunction |
| Resource Utilization | Poor | PiMfunction |
| Processes $\theta$. | Good | GaussianMfunction |
| **Input 3** | Excellent | PiMfunction |
| | Very Low | Zmfunction |
| Resilience $R_i$ | Low | Gauss2Mfunction |
| **Output** | Moderate | GbellMfunction |
| | High | PiMfunction |
| | Very High | PiMfunction |

**Table 3 Sample rules of the R-FIS 27 Rule Base** (Rules and weights to be determined by experts and/or stakeholders)

| Rules premise | Rules Consequence | Weight |
|---|---|---|
| If ($H$ is Low) & ($G$ is Very Low ) & ($\theta$ is Poor) THEN | (Resilience is very low) | 1 |
| If ($H$ is Low) & ($G$ is Low) & ($\theta$ is Excellent ) THEN | (Resilience is Low) | 0.8 |
| If ($H$ is Low) & ($G$ is High) & ($\theta$ is Excellent) THEN | (Resilience is Moderate) | 0.8 |
| If ($H$ is High) & ($G$ is High) & ($\theta$ is Excellent) THEN | (Resilience is Moderate) | 1 |
| If ($H$ is Very High) & ($G$ is Very Low) & ($\theta$ is Good) THEN | (Resilience is High) | 0.7 |
| If ($H$ is Very High) & ($G$ is High) & ($\theta$ is Good) THEN | (Resilience is High) | 1 |
| If ($H$ is Very High) & ($G$ is High) & ($\theta$ is Excellent ) THEN | (Resilience is Very High) | 1 |

**Table 4. Linguistic variables input template** (to be used with Table 1 as a scoring guide)

| Linguistic Variables Dimension | Tick the grey box next to your linguistic rating | Tick the grey box that best reflects your score of your linguistic rating | | | | | |
|---|---|---|---|---|---|---|---|
| Hazard Absorbing Capacity **(H)** | Low | | 1 | | 2 | | 3 | |
| | Moderate | | 4 | | 5 | | 6 | |
| | High | | 7 | | 8 | | | |
| | Very High | | 9 | | 10 | | | |
| Resource Availability **(G)** | Low | | 1 | | 2 | | 3 | |
| | Moderate | | 4 | | 5 | | 6 | |
| | High | | 7 | | 8 | | | |
| | Very High | | 9 | | 10 | | | |
| Resource Utilization Processes **(θ)** | Poor | | 1 | | 2 | | 3 | |
| | Good | | 4 | | 5 | | 6 | |
| | Very Good | | 7 | | 8 | | | |
| | Excellent | | 9 | | 10 | | | |

Location/city

Date of assessment

Assessors' name

**Table 5 Study locations- demographic and topographic summary** (Source: http://census.gov and United States Geological Survey Topographic Maps)

| | Windsor NC | Greenville  NC | Norfolk  VA |
|---|---|---|---|
| Location type | Small town | City | Large city |
| Types flood | River/storm/ rain | River /storm/ Rain | Coastal /river rain/storm |
| Total Population* | 3,630 | 84,554 | 242,803 |
| Male * (%) | 59.3 | 45.8 | 51.8 |
| Female* (%) | 40.7 | 54.2 | 48.2 |
| Median income * ($) | 29,063 | 34,435 | 44,480 |
| Poverty rate * (%) | 27.8 | 32.5 | 21 |
| Median Age*  (yr) | 38.6 | 26.0 | 29.7 |
| Under 14* (%) | 12.4 | 15.9 | 17.7 |
| 75 above* (%) | 8.7 | 4.3 | 4.6 |
| US Citizenship *(%) | 97.9 | 96.8 | 96.6 |
| Non English speaking *(%) | 5.83 | 6.74 | 10.3 |
| No of Households* | 1,088 | 36,071 | 85,485 |
| Family household* (%) | 61.2 | 46.3 | 58.7 |
| Average household size* | 2.29 | 2.18 | 2.43 |
| Household with individuals above 65* (%) | 34.1 | 14 | 20.3 |
| No of Housing units* | 1,193 | 40,564 | 95,018 |
| housing units occupied* (%) | 91.2 | 88.9 | 91.0 |
| Mean property Value ($)* | 93,800 | 147,100 | 193,400 |
| ** Elevation  (meter ) | 7.62 | 17.07 | 9.14 |

**Table 6. Input scoring and R-FIS resilience index output**

| Experts Scoring / Community | Model Input | | | | | | Model Output |
|---|---|---|---|---|---|---|---|
| | Hazard Absorbing Capacity (**H**) | | Resource Availability (**G**) | | Resource Utilization Processes ($\theta$) | | Resilience Index **R** |
| | Linguistic Score | Score | Linguistic Score | Score | Linguistic Score | Score | |
| Norfolk, VA | High | 7.0 | High | 8.0 | Good | 6.0 | **0.836** |
| Greenville, NC | High | 8.0 | Moderate | 6.0 | Very Good | 8.0 | **0.9** |
| Windsor, NC | Moderate | 4.0 | Low | 2.0 | Very Good | 8.0 | **0.477** |

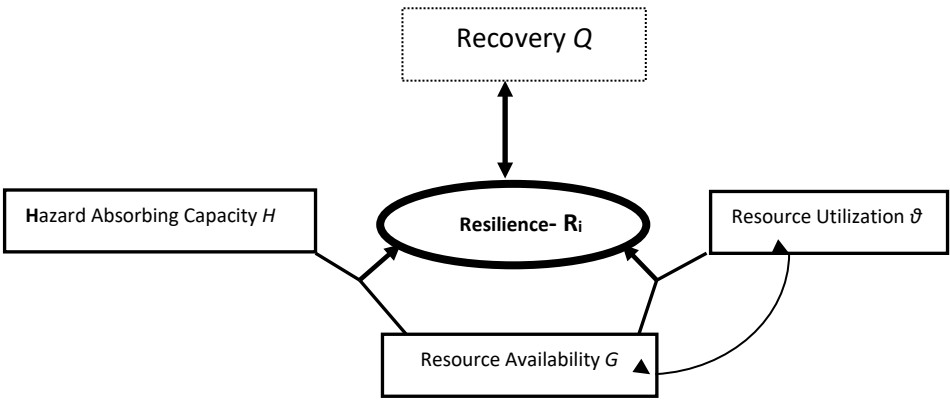

**Figure 1.** Resilience measuring conceptual framework

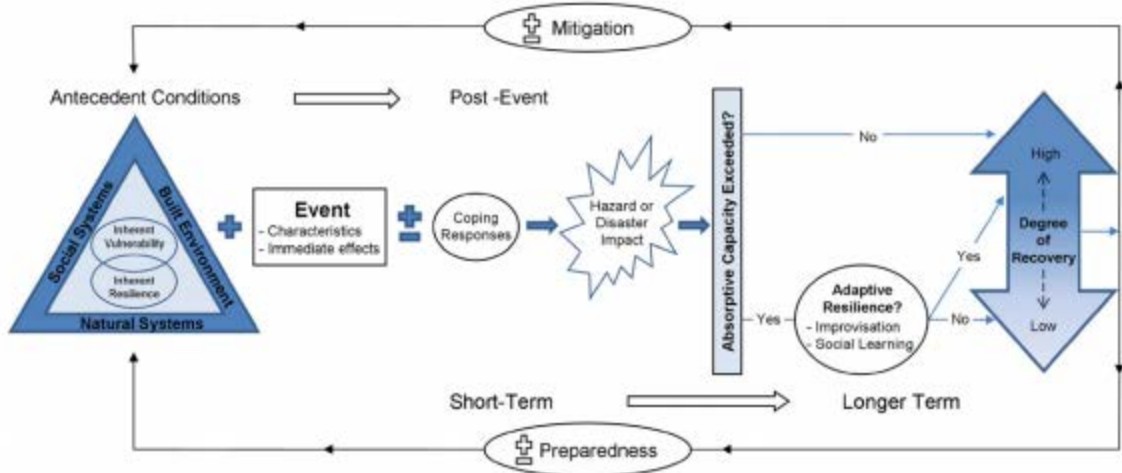

Schematic representation of the disaster resilience of place (DROP) model.

Figure 2: The Disaster Resilience of Place (DROP) model reproduced from Cutter et al, (2008). A place-based model for understanding community resilience to natural disasters. This model illustrates the interelationship between resilience and recovery within the hazard–resilience system.

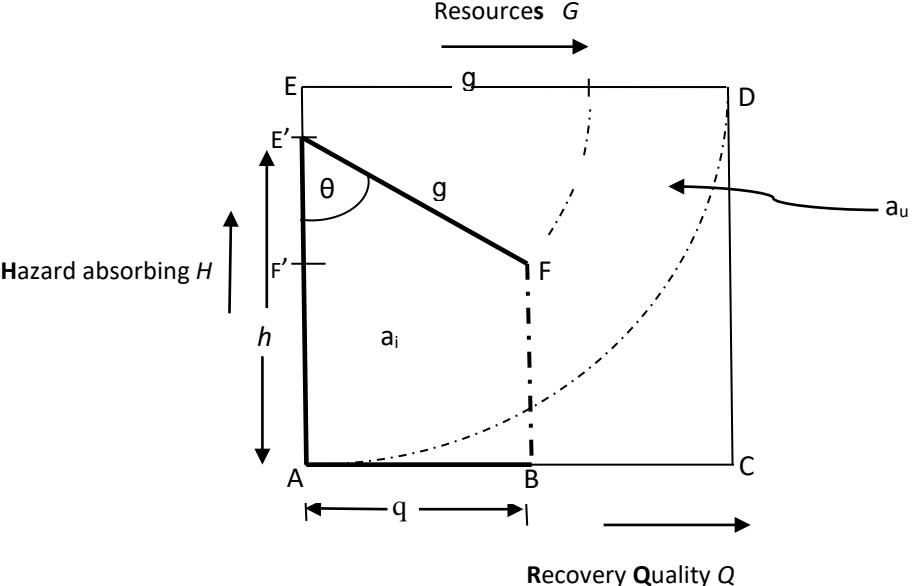

**Figure 3:** Resilience conceptual model. A geometric model used to derive appropriate mathematical relationships of the proposed framework and provide some insights

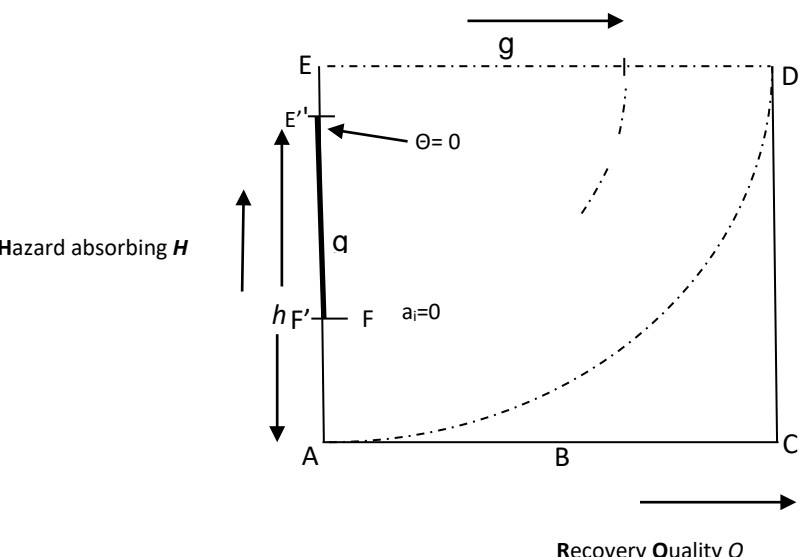

**Figure 4.** Resilience area = 0 when ρ= Sin Θ= 0. A variation of model Figure 3 depicting an extreme case of a community with zero efficiency in resource utilization.

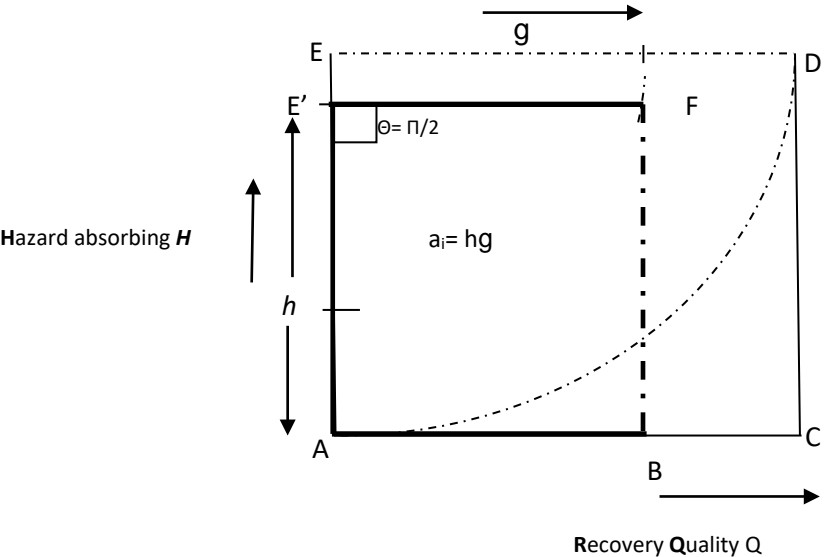

**Figure 5.** Resilience area ($a_i = hg$). A variation of model Figure 3 depicting an extreme case of a community with a perfect resource utilization system (efficiency of 1.0) which maximizes recovery resources' g on absorbing capacity h.

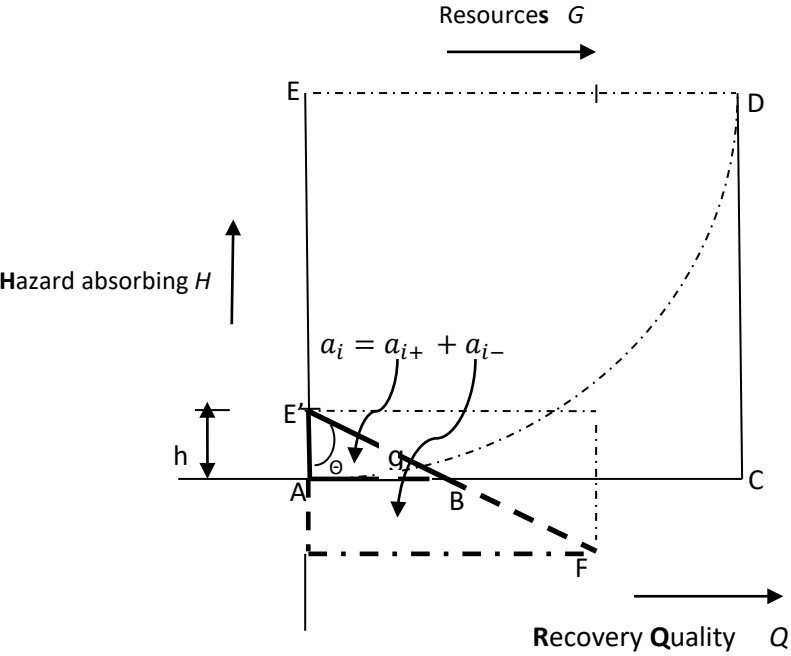

**Figure 6.** Resilience as absorbing capacity approaches zero

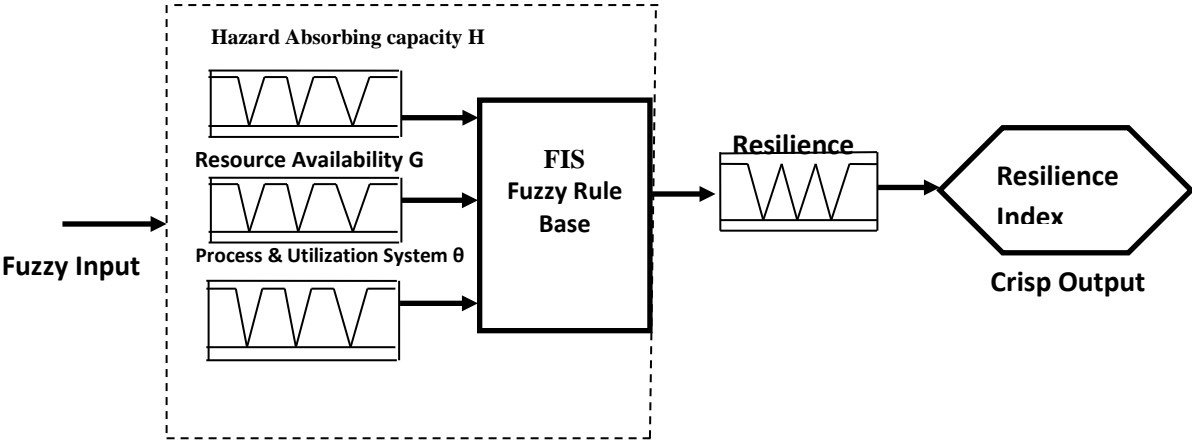

**Figure 7.** Resilience fuzzy inference systems

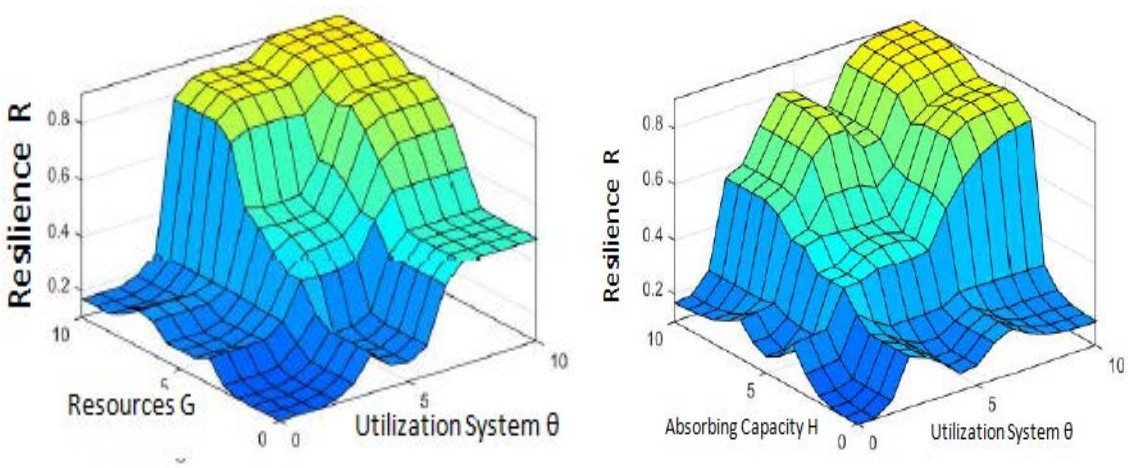

**Figure 8.** Examples of resilience output surface plots.

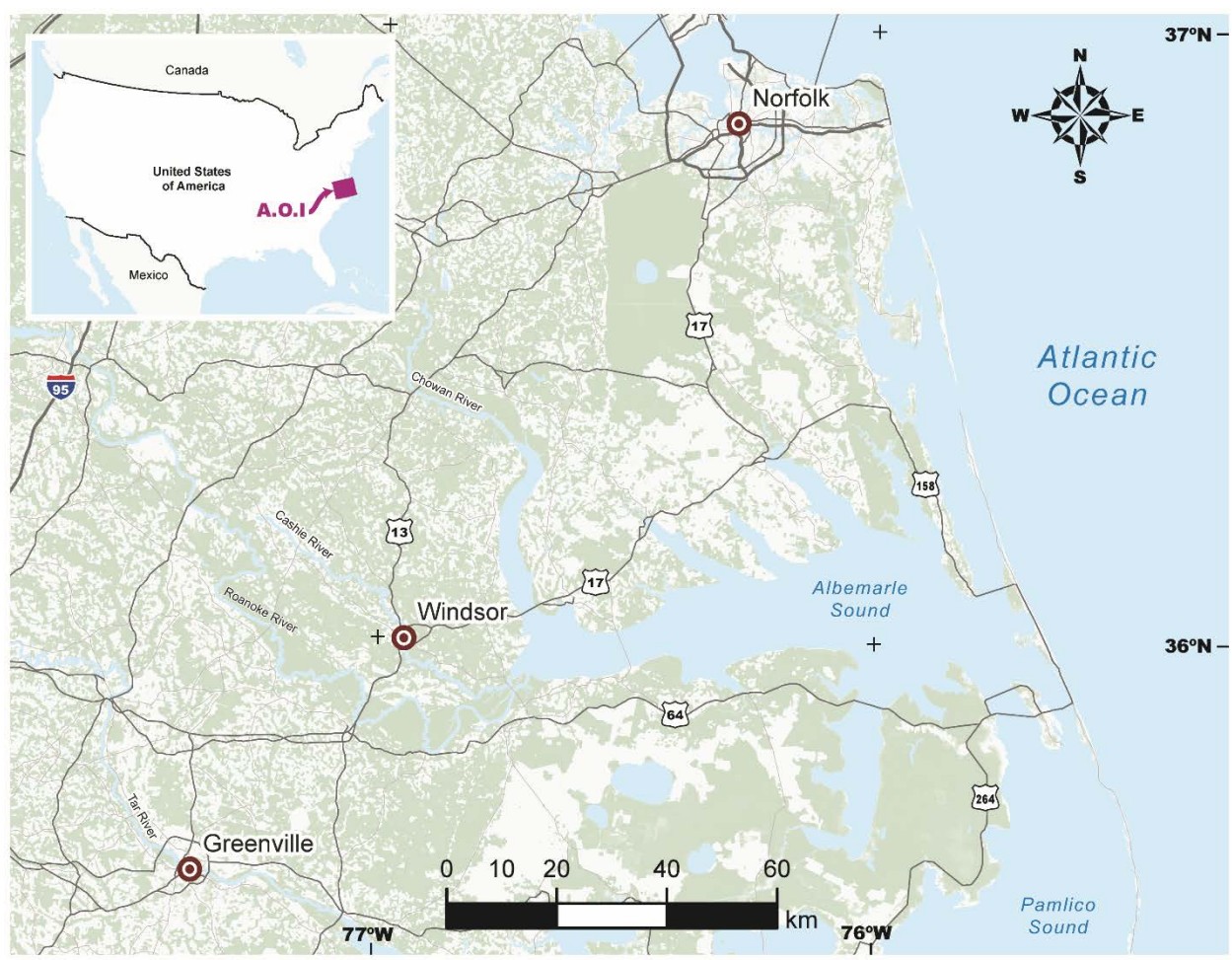

**Figure 9.** The study area on map showing Greenville, NC; Windsor, NC and Norfolk VA
Source: Produced in the GIScience Center, East Carolina University

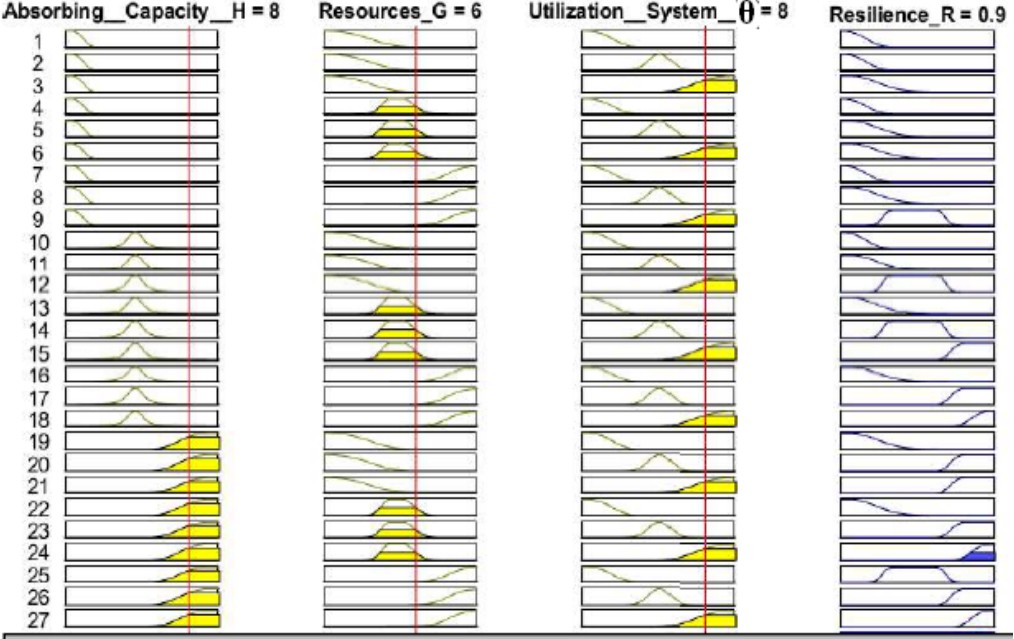

**Figure 10.** Rule setting and output for Greenville

🟨 Active input membership functions 🟦 Active output membership function

1    Towards Measuring Resilience of Flood Prone Communities: A Conceptual Framework

2                              V.O.Oladokun[1] and B.E.Montz[2]
3          [1]Department of Industrial and Production Engineering, University of Ibadan, Ibadan, Nigeria
4          [2]Department of Geography, Planning, & Environment, East Carolina University, Greenville, NC, USA

7                                            **Abstract**
Community resilience has become an important policy and research concept for understanding and
addressing the challenges associated with the interplay of climate change, urbanization, population
growth, land use, sustainability, vulnerability and increased frequency of extreme flooding.
Although measuring resilience has been identified as a fundamental step toward its understanding
and effective management, there is, however, lack of an operational measurement framework due
to the difficulty of systematically integrating socio-economic and techno-ecological factors. The
study examines the challenges, constraints and construct ramifications that have complicated the
development of an operational framework for measuring resilience of flood prone communities.
Among others, the study highlights the issues of    proliferation of definitions and conceptual
frameworks of resilience, challenges of data availability, data variability and data compatibility.
Adopting the National Academies' definition of resilience, a conceptual and mathematical model
was developed using the dimensions, quantities and relationships established by the definition. A
fuzzy logic equivalent of the model was implemented to generate resilience indices for three flood
prone communities in the US. The results indicate that the proposed framework offers a viable
approach for measuring community flood resilience even when there is a limitation on data
availability and compatibility.
Keywords: Hazard, Disaster, Flood, Resilience, Measurement, Fuzzy, Community

**1.0 Introduction**

Developing resilience of communities has become widely recognized as critical for disaster risk management due to the increased incidents of extreme weather events, such as flooding, which have disrupted economic activities, caused huge losses, displaced people and threatened the sustainability of communities across the world (Cai et al., 2018; Cutter 2018; Mallakpour and Villarini, 2015; Montz, 2009; Oladokun et al., 2017; Su, 2016a; Wing et al., 2018). Major international policy instruments such as the United Nations International Strategy for Disaster Reduction's (UNISDR) 2015 Strategic Framework and the 2005 Hyogo Framework have emphasized and adopted resilience principles in disaster risk management (Cai et al., 2018; Cutter et al., 2016). For instance, the interplay of extreme floods, population growth and rapid urbanization has increased flood hazard risks such that conventional flood risk management (FRM) measures of concrete structures, levees, flood walls and other defenses have become inadequate and unsustainable across various communities (Duy et al., 2018; Guo et al., 2018; Trogrlić et al., 2018; Wing et al., 2018). Resilience has gained a lot of attention, from both policy and research perspectives, involving using it to understand and address the challenges of land use, vulnerability and sustainability in the context of flooding (Cohen et al., 2016; Cohen et al., 2017; Folke, 2006; Parsons et al., 2016; Sharifi, 2016). Building community resilience has emerged as particularly relevant in dealing with flooding, which has become the most widespread and destructive of all natural hazards globally (Jha et al., 2012; Mallakpour and Villarini, 2015; Montz, 2009).

Consequently, there has been a shift from relying solely on large-scale flood defense and structural systems towards an approach that emphasizes the concept of community resilience as a strategic component of flood risk management (Hammond et al., 2015; Park et al., 2013). This shift is being reinforced by a consensus that since floods cannot be all together prevented; FRM must focus more on building the resilience of flood prone communities (Joseph et al., 2014; Oladokun et al., 2017; Schelfaut et al., 2011).

There is a consensus that the first and fundamental step toward understanding and operationalizing resilience for flood disaster and hazard management is to have an acceptable resilience measuring template (NRC, 2012). For instance, the ability to understand and objectively evaluate the impact of FRM programs, interventions and practices on community flood resilience is needed for making

political and business cases for proactive FRM investment from both public and private sectors.
Cutter (2018) suggested that an acceptable template is a basic foundation for monitoring baselines
and progress in building hazard resilience.
Furthermore, a measuring template will be useful as a decision support tool for the efficient
deployment of scarce FRM resources and also provides a basis for monitoring resilience changes
with respect to resource deployment.  For instance, Keating et al. (2017) explained that there is a
need for the continued development of theoretically sound, empirically verified, and applicable
frameworks and tools that help in understanding key components of resilience in order to better
target resilience-enhancing initiatives and evaluate the changes in resilience as a result of different
capacities, actions and hazards.
Therefore, the search for an acceptable framework and empirical model for measuring resilience
remains relevant and continues to attract attention (Cutter et al., 2016; Zou et al., 2018;  Cai et al.,
2018; Keating et al., 2017). Some existing measuring approaches, as identified in Cai et al., (2018),
include the Baseline Resilience Indicators for Communities (BRIC), the Resilience Inference
Measurement (RIM) framework, the National Oceanic and Atmospheric Administration (NOAA
2010) Coastal Resilience Index, the PEOPLES Resilience Framework, and the Communities
Advancing Resilience Toolkit (CART). There is also the '5C-4R' Zurich Alliance framework
combining the 'five capitals' of the UK's Department for International Development sustainable
livelihoods framework (Scoones, 1998) and the four properties of a resilient system (Szoenyi, et
al., 2016): the framework incorporates a technical risk grading standard (TRGS) developed by
Zurich risk experts  (Keating et al. 2017).
Despite the attention resilience has gained, the concept remains difficult to operationalize in the
context of community flood risk management due to, among other factors, the difficulty in
measuring resilience (Cutter, 2018; Fisher, 2015). Many experts and authors have noted  the
difficulty in integrating indicators of the natural and human systems as well as socio-environmental
factors into resilience by most of the existing frameworks (Cai et al., 2018; Cutter, 2018; Fuchs
and Thaler, 2018; Qiang and Lam, 2016).  Resilience, as a multifaceted and multidimensional
concept, has developed across multiple disciplines and applications such that resilience discourse
has attracted multidisciplinary interests from both research and policy perspectives.  While the
wide spectrum of multidisciplinary and practice interests characterizing resilience discourse has
increased its understanding and generated insights, it has also led to the emergence of multiple
variants of its definiton as well as the absence of consensus on the conceptual framework for its
measurement (Brown and Williams, 2015; Cohen et al., 2016; Cutter 2018). For instance,
resilience has been noted to have varied definitions depending on the hazard and disciplinary
contexts, with over 70 definitions identified by Fisher (2015).
The multiplicity of definitions has led to proliferation of conceptual models, frameworks and
interpretations (Costache, 2017), such that there is difficulty in transforming resilience
measurement from an abstract concept into an objective operational quantitative template.
According to Cutter (2018), the difficulties in harmonizing and operationalizing these definitions
have led to the emergence of a wide array of measurement approaches. Meanwhile, a pre-requisite
to having an operational model, in the context of resilience measurement, is the adoption or
convergence of definition by the resilience research and policy community. Such a definition
should meet the following criteria: i) emanates from or receives the formal endorsement of a
widely recognized institutional platform of stakeholders, ii) encompasses a wide spectrum of
existing resilience concepts, iii) has some degree of simplicity, and iv) enjoys high acceptance of
both the research and policy community. In a widely cited National Research Council report
(NRC, 2012), the US National Academy of Sciences defines resilience as the ability of a system
to prepare and plan for, absorb, recover from, and more successfully adapt to adverse events (Cai
et al., 2018; Cutter, 2018). Therefore, this study has adopted this definition as the basis for the
proposed framework for measuring the resilience of flood prone communities.
From a systems perspective, community-resilience is a non linear collection of socio-ecological,
socio-political, techno-ecological and socio-economic entities, each characterized by dynamic and
complex spatiotemporal interactions. Essentially, the concept of resilience involves the
interactions of several entities each defined by some social, economic, natural, technical and
environmental dimensions (Cai, et al., 2018; Norris et al., 2008). For instance, the community
component was succinctly described by Cai et al. (2018) as a coupled natural and human system
that manifests various sources of complexity such as nonlinearity, feedback, and uncertainty and
dynamic interactions.
Furthermore, coupled with the challenge of complexity and the dynamic nature of community-
resilience modeling is the challenge of data and computational analysis. It has been established
that information and data items characterizing community-resilience system are mostly imprecise,
incomplete, vague, complex, fuzzy and subjective within the context of flood risk management
(Kotze and Reyers, 2016; Oladokun,et al., 2017). These characteristics present some operational
and analytical challenges for any complex model based on traditional crisp mathematics and hard
computational approaches   because of data availability, data variability and data compatibility.
The resilience measuring problem with its interplay of definitional ambiguities, multi-
dimensionality, and spatiotemporal dynamics invariably results in complex mathematical models.
Such models, given the level of incompleteness, vagueness, and subjectivity that characterizes the
human and socio-political aspects of resilience, offer little tractability with conventional hard
computational tools and are difficult to operationalize. Hence, Oladokun et al. (2017) suggested
that a resilience measuring model may be more amenable to a soft computing analytical technique
such as fuzzy logic.
**1.1 Aim and objectives**
Based on the background presented above, this study is aimed at adopting a soft computing
approach, a fuzzy logic computational model, for the proposed flood resilience measuring
template. In particular, the objectives of the study are 1) the development of a descriptive model
that outlines our abstract interpretation of community resilience as a system, using insights from
relevant literature, interactions with experts and observations of selected flood prone
communities, 2) development of an equivalent mathematical model of the resulting descriptive
model using an appropriate tool to generate further insights, and 3) development of an equivalent
fuzzy inference system suitable for computational and analytical purposes in the face of the
aforementioned data issues. The next section briefly describes some relevant fuzzy logic concepts.
**1.2 An Overview of Fuzzy Logic**
Fuzzy set theory provides a mathematical tool for modeling uncertain, imprecise, vague and
subjective data which represents a huge class of data encountered in most real-life situations
(Adnan et al., 2015; Lincy and John, 2016). The fuzzy logic (FL) concept, introduced in 1965 by
Lot A. Zadeh, is an extension of the classical set theory of crisp sets**.** FL, like humans,
accommodates grey areas where some questions may not have a clear Yes or No answer or black
and white categorization. According to Zadeh (1996), Fuzzy Logic = Computing with Words. FL
mimics human reasoning and capability to summarize data and focus on decision-relevant
information in problems involving incomplete, vague, imprecise or subjective information. It is a
computational concept that allows for modeling of complex systems using a higher level of
abstraction originating from our knowledge and experience. It provides a very powerful tool for
dealing quickly and efficiently with imprecision and nonlinearity (Oladokun and Emmanuel,
2014). This capability to mine expert knowledge and use limited or fuzzy data makes fuzzy
inference systems (FIS) a suitable tool for resilience measurement modeling.
The concept of membership function (MF) is central to FIS. In traditional logic, an element $x$ is
either in or out of crisp set A; in other words, its degree of membership of the set is either zero or
one.   However, in fuzzy logic the element $x$ can be in a fuzzy set B 'partially' by using a MF
$\mu_B(x)$ which can return any real value between 0 and 1. This returned value is the degree of
membership representing the degree to which the element belongs to a fuzzy set. Therefore, in FL,
the truth of any statement becomes a matter of degree.
Thus for crisp set A   $\mu_A(x) = \begin{cases} 1 & if\ x \in A \\ 0 & otherwise \end{cases}$
On the other hand, for a fuzzy set, the MF may be represented as follows
$\mu_B(x) = \begin{cases} f(x) & if\ b_1 \le x \le b_2 \\ g(x) & if\ b_2 < x \le b_3 \\ 0 & otherwise \end{cases}$
Actually, the crisp set is a special case fuzzy set whose MF returns only zero or one. There are
many functions that are used as MFs. Some widely used MFs, Generalized bell shaped, Gaussian
curves, Polynomial curves, Trapezoidal, Triangular and Sigmoid MFs (Oladokun and Emmanuel,
2014; Adnan et al., 2015).  The Mamdani FIS approach (Mamdani and Assilian, 1975), adopted
for this study, is made up of a fuzzy inference engine characterized by the use of carefully selected
MFs and a fuzzy rule base. The rule base is a set of 'IF THEN' statements that capture experts'
knowledge of the logic governing the problem.  The fuzzy inference system will provide a template
for experts and other stakeholders to translate their perceptions of the problem and map their
linguistics rating of these variables   into a resilience index based on the fuzzy relationships we
define.

**2.0 Resilience Measuring:  A Conceptual Framework**

**2.1 Descriptive model**

The design objective is to have a conceptual framework and its associated mathematical model with sufficient tractability by minimizing the number of model elements and adopting the barest minimum relationships while maintaining a reasonable level of validity. Therefore, as the theoretical basis for the proposed conceptual model, as mentioned earlier, we are adopting the resilience definition put forward by the US National Academies (NRC 2012). Conceptually this definition implies that a community's resilience is a quantity that reflects capacities such as: 1) the community's coping capacities, in terms of a threshold of hazard it can absorb (Hazard Absorption Capacity H), 2) its accessible resources (Resource Availability G), and 3) its resource utilization efficiency determined by factors like its preparedness and its governance processes (Resource Utilization Processes $\theta$). These capacities interact to define its ability to prepare for, absorb, recover from, and more successfully adapt to adverse flooding events. We attempt to conceptualize this understanding as shown in Figure 1.

Each of the dimensions in Figure 1 is influenced by a number of technical, social, ecological, economic, and political factors following work that has been reported in the literature which sheds light on these factors and how they influence the dimensions (see Cohen et al., 2016; Lee et al., 2013; Rose, 2017). For example, hazard absorbing capacity H is determined by a number of techno-ecological factors such as adequacy, sophistication and use of infrastructure and technology as well as redundant capacities. It is also determined by socio-ecological and socioeconomic factors that influence both individual and institutional coping capacities. Resource availability is determined by things like community capital, political influence, and economic activities as well as ecological resources accessible to drive the quality and timeliness of recovery. Resource utilization processes are determined by the quality of governance and institutions such as judiciary, police, media, and public service. These processes influence policy formulation and implementation, the ease of doing business and the efficiency of use of resources. A detailed structured and operational rendition of the foregoing is presented in sections 2.2 and 3.3.

Figure 1 here

Furthermore, in the context of FRM, the framework of Figure 1 recognizes that resilience enhances
recovery or that recovery is an outcome of resilience whereby when a community, as a coupled
system, becomes more resilient its capacity to experience post disaster recovery increases. In other
words, recovery, in terms of time taken to attain post disaster recovery and the degree of recovery
attained, is influenced by its resilience. Invariably the conceptual framework implicitly suggests
that recovery (recovery speed and recovery quality) can surrogate resilience. This is reasonable
because post disaster recovery is driven by resilience factors such as preparedness, and coping
capacity, among others. This understanding is supported by the DROP disaster resilience model
of place (DROP) as illustrated in Cutter et al. (2008), reproduced in Figure 2.
Figure 2 here
**2.2 Mathematical model**
The next stage is to transform the conceptual framework of Figure 1 into an operational
mathematical model. This is accomplished by defining a geometric model of the framework as
shown in Figure 3. This model is then used to derive appropriate mathematical relationships for
resilience measurement and provide some insights.
**2.2.1 Notations, definitions and terms**
We adopt the following notations, definitions and terms   to explain the components of Figure 3 in
the context of flood hazard.
i.   Hazard Absorbing Capacity ($H$): ($H=h$: $0 \leq h \leq 1.0$). The resilience of a community
depends on the level of the flood hazard the community systems can absorb before
totally collapsing or undergoing irreversible disintegration. $H=1$ is the highest
absorbing capacity whereby the community can absorb and survive the damages and
disturbance (both structural and non structural) of the most severe category of flooding
conceivable. This captures various resilience factors such as coping capacity,
redundancy, preparedness, sense of place attachment and other capacities as explained
in Table 1.
ii.  Resource Availability ($G$).  This is the quantum of resources available to plan and
pursue recovery as well as achieve recovery quality level $Q$ (including adaptive
recovery). Note that $G=g$ ($0 \leq g \leq 1.0$) captures both economic and community capital.
It is the measure of resources the community is able to attract as a result of its overall

| 233 | | economic and political influence, its natural assets, and human capital assets (see Table |
|---|---|---|
| 234 | | 1 for further details). |
| 235 | iii. | Resource Utilization Processes ($\theta$): With $0 \leq \theta \leq \Pi/2$, we define $\rho$ ($\rho = \mathrm{Sin}\ \theta$) as system |
| 236 | | efficiency. This is a resilience component that affects recovery and revolves around |
| 237 | | factors such as preparedness, community governance, institutional systems and |
| 238 | | processes. It determines the efficiency and effectiveness of the use of resources to |
| 239 | | achieve  recovery and establish adaptive capacity. In other words, how *well* resources |
| 240 | | are used is as important as how *much* of a set of resources is used in building resilience. |
| 241 | | It measures the probity, level of accountability, level of waste, corruption, red-tapism, |
| 242 | | and bureaucracies within the system. A community with strong institutions such as a |
| 243 | | functioning judiciary and an efficient civil service, for instance, will tend to return high |
| 244 | | $\rho$. So an ideal or utopian community will have its $G$ deployed at $\theta = \Pi/2$, such that  $\rho =$ |
| 245 | | $\mathrm{Sin}\ (\theta) = \mathrm{Sin}\ (\Pi/2) = 1$. |
| 246 | iv. | Recovery Quality Level ($Q$).  This represents the outcome of post hazard conditions in |
| 247 | | terms of restoration quality and socio-ecological functionality, among others. |

| 248 | | The following definitions apply with reference to Figure 3 |
|---|---|---|
| 249 | v. | $a_i$ :  Resilience reservoir of a real system i is defined as the  area of trapezium ABFE' |
| 250 | | determined by  the hazard absorbing capacity,  at $H = h$, of the system, the available |
| 251 | | quantum of resources ($G = g$), the quality of governance processes and resource |
| 252 | | utilization systems ($\mathrm{Sin}\ \theta$) and the achievable  recovery quality ($Q = q$) |
| 253 | vi. | $a_u$ : The resilience reservoir of an utopian (ideal)  system is defined as the area of square |
| 254 | | ACDE. This occurs at ideal FRM conditions: that is, a community system with |
| 255 | | adequate resources, perfect governance and processes with zero waste of resources and |
| 256 | | infinite hazard coping threshold when h= AE (or at maximum absorbing capacity), |
| 257 | | g=ED (maximum resource adequacy) and $\theta = \Pi/2$ (perfect or utopian system with |
| 258 | | 100% efficiency or $\mathrm{Sin}\ \theta = 1.0$). The utopian system can achieve a perfect recovery |
| 259 | | index Q= q= 1.0 or Q=AC |

| 260 | Extensive review of the literature was carried out to provide an informed basis for mapping |
|---|---|
| 261 | FRM factors and inputs to the dimensions of resilience.  This is summarized as shown in Table |

1. Theoretically, the values of the dimensions H, G, θ can be estimated from adequate data on
these input factors and appropriate functions.
**Table 1 here**
Figure 3 here
**2.2.2 Resilience modeling**
The utopian resilience reservoir is the benchmark for evaluating resilience such that actual
resilience $R_i$ can be defined as the ratio of $a_i$ to $a_u$ as indicated in equation 1.
$R_i = \dfrac{a_i}{a_u}$     (1)
Using the insights from Figure 1, we attempt to develop the mathematical model implied in
equation 1 (note R is dimensionless since both $a_i$ and $a_u$ are areas).
$a_i = \frac{1}{2}\{AE' + BF\}AB$     (2)
$a_u = AE \times ED$
$a_u = H \cdot G$     (3)
Note:  $AE' \equiv h$     (4)
$BF = AE' - F'E' = h - gCos\theta$     (5)
$AB = F'F = gSin\theta$     (6)
Putting 4, 5, 6 into 2
$\Rightarrow a_i = \dfrac{1}{2}\{h + (h - gCos\theta)\}gSin\theta$
$a_i = hgSin\theta - \frac{1}{2}g^2 Sin\theta Cos\theta$
$a_i = hgSin\theta - \frac{1}{2}g^2 Sin\theta \pm \sqrt{1 - Sin^2\theta}$
Recall we define 'Efficiency of resource utilization system' as ρ =Sinθ
$\therefore a_i = hg\rho - \frac{1}{2}g^2\rho\sqrt{(1 - \rho^2)}$     (7)
Putting 3 and 7 into 1
$$R_i = \frac{hg\rho - \frac{1}{2}g^2\rho\sqrt{(1-\rho^2)}}{HG} - \qquad (8)$$
Without loss of generality, h and g are treated as indices such that
$0 \leq h \leq 1 \quad and \; 0 \leq g \leq 1$
Then $H=G=1$ in equation 8 which implies
$$R_i = hg\rho - \frac{1}{2}g^2\rho\sqrt{(1-\rho^2)} \qquad (9)$$
Equation 9 is a valid expression for resilience.
That is, $R_i = f(h,g,\rho)$,
Where h, g and h are as explained in section 2.2.1 and their values are decided by experts and/or
stakeholders, varying depending upon the location and scale of application of the model.
**2.2.3 Some insights from model using some extreme values**
This section discusses some example cases of the model (equation 9) output using selected
hypothetical extreme parameters' values  to generate further insights into model structure (with
reference to Figure 1). The 'extreme' scenarios analysis is used to demonstrate how each of the
three  dimensions impacts R.
**Case 1:  As  $\rho \to 0$     $R \to 0$**
In fact, R= 0 when $\rho = 0$. This may be interpreted as the case when the resource utilization
processes have zero efficiency (see Figure 4) or a collapsed governance system such as when a
flood disaster occurs in a community ravaged by civil war with breakdown of law and order. In
such situations, community resilience is nil as all resources put into recovery will be 'wasted,'
irrespective of the level of coping or infrastructure previously in place.

Figure 4 here

**Case 2: As  $\rho \to 1$    $R \to hg$**
This implies that θ=Π/2 or Sinθ=1 which depicts an ideal situation when the communal processes,
FRM resource administration, and utilization systems are highly efficient and near perfect.  Under

this scenario, the resources $g$ and community's coping capacities contribute maximally to resilience (see Figure 5).

Figure 5 here

**Case 3: $g \to 0$     $R_i \to 0$**     Resilience disappears when resources dry up.

**Case 4: h= 1**   Resilience is determined by resource availability and utilization

**Case 5:   As   $h \to 0$     $R \to 0^-$**

From Figure 6, resilience approaches zero from negative reservoir quadrant when h=0 (i.e. coping and absorbing capacities disappear or collapse) and $\rho < 1$ (efficiencies of resource use, preparedness, and governance systems fall below 1). The 'Negative' resilience reservoir quadrant characterizes vulnerable communities. Note that vulnerability is sometimes seen as the flip side of resilience (Folke et al., 2002) or a complementary community-hazard management concept (Cutter, 2018; Fekete and Montz, 2018; Shah et al., 2018). Hence from figure 6 as the absorbing/coping capacity h approaches zero, a community enters vulnerability mode because more resilience area lies below the positive plane. In other words, equation 9 suggests that a community without coping or built in absorbing capacities is vulnerable, especially if its governance structure is poor (i.e. $\text{Sin}\theta \to 0$).

Figure 6 here

**3.0 Resilience fuzzy inference system (R-FIS):  Computer model**

While the resulting model of equation 9 provides useful insights, its application however is premised on the availability of clear information on input factors and adequate data for estimating model parameters, That is, complete data as described in section 2.2 and Table 1, for estimating dimensions H, G and θ.  However, there are issues of data availability and data compatibility (Parsons et al., 2016) which make it inefficient to do crisp estimation of these parameters. Therefore, to operationalize the proposed framework, a (FIS) equivalent has been developed.

A computer model of the proposed R-FIS (Figure 7) was designed in the Matlab fuzzy logic
development environment. The environment was adopted because it supports easy to use graphical
user interface (GUI) tools and has multiple MFs for implementing a FIS. A process consisting of
systematic review of the literature, interactions with experts, meetings with community leaders,
interviews of other stakeholders and field observations (described in more detail in Section 4.1)
was used to gain insights for specifying the R-FIS's design and inference engine's elements (Table
2) as well as determine appropriate IF THEN statements for the rule base (Table 3). With three
input linguistic variables, each with three term sets (or possible values), there can be up to 27
explicit input variable combinations, or 27 explicit fuzzy rules combinations.  Table 3 is a sample
extract from the 27 'IF THEN' statements of the rule base.
Figure 7 here
Table 2 here
Table 3 here
Figure 8 shows the 3D surface plot resulting from an infinite combination of input factors.  The
shape of the resilience surface is determined by the rules (Table 3) and the selected membership
functions (Table 2) used to express the term sets. This shape can be varied by modifying the
membership functions, the term sets, the rules and their weights to reflect new realities and
understandings about the resilience systems. This gives flexibility to simulate various
combinations of parameters in order to arrive at an optimum design.


Figure 8 here

**3.2. Model expert scoring framework**

Although information and explanations in Table 1, in principle, give a general guide for evaluating and quantifying these dimensional inputs of the resilience model, there is still the need for an easy to use operational template for capturing experts' input into the FIS in relatively standardized fashion. Table 4 is an example of such an input template designed for this study. A typical application procedure is described in section 4.1 with the case study communities.

**Table 4 here**

**4.0 Model Application: Study location**

The following describes the application of the model using three flood prone communities in the United State (U.S.). Following decades of experience in dealing with hazards and disasters, cities and institutions in the U.S. offer considerable information and insights in community resilience systems management (Su, 2016b). Two coastal states of North Carolina and Virginia are home to many flood prone communities of various sizes with diverse socio-economic and techno-ecological characteristics that readily lend themselves to a study of resilience. Both states have adopted a number of FRM programs, policies, and strategies for building flood resilience across many rural and urban communities (North Carolina Floodplain Mapping Program, 2019; Mogollón et al., 2016). Specifically, Norfolk, VA a coastal city in Virginia with a massive naval base, Greenville, NC, a large university town, and Windsor, NC a small riverine rural town were selected (Figure 9). Table 5 summarizes some vital socio- economic features of these communities.

Figure 9 here

Norfolk, located on the Chesapeake Bay and near several rivers, experiences precipitation flooding, when the intensity of rainfall exceeds stormwater drainage capacity, storm flooding from hurricanes and nor'easters, and tidal flooding due to its elevation and coastal location. Greenville, with relatively flat topography is located on the Tar River and is traversed by a number of small streams (Pitt County Development Commission, 2019). Besides riverine flooding, the relatively flat topography of its coastal plain location leads to flooding from intense or long-lasting rain events such that the stormwater system is incapable of handling the overland flow. Located on the

meandering Cashie River in eastern North Carolina, Windsor has experienced four major floods
since 1999, all from tropical storms. Thus, not only are the communities different demographically,
but they have rather different flood regimes and histories, with Windsor and Greenville
experiencing riverine flooding, though with very different patterns of damage, and Norfolk
experiencing a combination of coastal and riverine flooding.

Table 5 here
**4.1 Model application: data gathering and results**
For the purpose of illustration, input scores were developed using the template shown in Table 4
along with the guidelines in Table 1 and the communities' information, summarized in Table 5.
The sample input data were generated based on the outcome of field studies and reflective
interactions with experts and stakeholders familiar with the study locations; these stakeholders
include academics, government officials and community leaders. In particular the sample scoring
was based on the insights derived from our understanding of their opinions, as well as demographic
and socio-economic information extracted from various historical and government records,
including the US census (Pitt County Development Commission, 2019; North Carolina Floodplain
Mapping Program, 2019; Mogollón et al., 2016). For instance, during a 2018 workshop by the
North Carolina Chapter of the American Planning Association held at Windsor, NC, the authors
had the opportunity to interact with and mine the knowledge of academics, students, city managers,
community leaders, relevant officials from emergency agencies, and curators of landmark centers,
among others. The authors also took tours of Norfolk, VA and Greenville, NC, under the guidance
of academics, GIS and FRM experts from the cities' universities.  These interactions and the
associated field studies provided insights for generating the sample scoring; the studies involved
interviews and qualitative assessment from site observations of community flood control projects
and individual property FRM retrofit systems. As an example, the perceptions of resident planning
experts and other stakeholders on how some ongoing flood risk management interventions would
have impacted the capacity of the community to cope with varying flood levels was useful in
classifying Hazard Absorbing Capacity, as was the extent and type of flood control and retrofit
projects.
Table 6 shows the results. Norfolk and Greenville both have relatively high hazard absorbing
capacities, with Norfolk rated as slightly lower owing to problems associated with the disruption
that regularly occurs from overland flooding combined with tidal flooding. Windsor's is lower
than Norfolk and Greenville but still moderate because of how the community has adapted to its
flood risk. Not surprisingly, Norfolk has the highest resource availability and Windsor the lowest
based on their size and relative wealth. At the same time, for the illustrative purposes here, size
and diversity of the communities are seen to be inversely related to resource utilization processes.
The model output, Resilience Index R, indicates that, based on the input values, Grenville's
resilience is slightly greater than Norfolk's while, not surprisingly, Windsor lags rather far behind.

Table 6 here
The input to output mapping implemented in Matlab fuzzy toolbox allows for infinite
combinations of input factors either by sliding or inputting the respective input variable axis on
the fuzzy rule interface. Figure 10 is a snapshot of the input combinations for Greenville, using the
scores from Table 6. The vertical bar (red line on each) can be moved to indicate how resilience
changes with a change in one or another (or all) of the three variables. The yellow shapes indicate
the rules (see the subset in Table 2) that contribute to each variable's score. All of the output, in
both Table 6 and Figure 8, is based on expert insights and understandings and thus provides a
dynamic template to measure resilience under different conditions. The proposed framework
accommodates the understanding that community resilience should be treated as a multifaceted
and multidimensional construct that can only be achieved by focusing on all aspects of a
community system. While the fuzzy implementation of the  framework can be used both as a
resilience index tool and  a resilience classification scheme, it is however, like many existing
resilience measuring models, still dependent on the subjective opinions of experts and other
stakeholders.
Figure 10 here
**5.0 Discussion and Conclusions**
Many previous studies have identified the multiplicity of definitions as one of the major
difficulties in transforming resilience measurement from an abstract concept into an objective
operational framework (Costache, 2017; Fisher, 2015; Oladokun et al., 2017). This study proposes
three criteria for adopting a suitable definitional basis for a framework conceptualization. These
criteria which address issues such as the need to achieve model simplicity and accommodate the
multidimensional nature of resilience (Brown and Williams, 2015; Cohen et al., 2016; Cutter 2018)
were used to recommend the National Academies' definition of resilience (NRC, 2012) as a robust
and viable basis for developing a measurement model.
Similarly, many scholars have highlighted dealing with the complexity involved in the integration
of indicators of natural and human systems into a community resilience model (Cai et al., 2018;
Cutter, 2018; Fuchs and Thaler, 2018; Qiang and Lam, 2016) as a key to transforming resilience
measurement from an abstract concept into an objective operational framework. To that end, we
adopt a three-component system in a way that reflects key relationships among technical, social,
ecological, economic, and political factors that have been reported in literature (Cohen et al., 2016;
Lee et al., 2013; Rose, 2017) as key to the multidimensional treatment of resilience.
Transforming the conceptual model into a quantitative template requires some sound theoretically
basis, a condition noted in Keating et al., (2017) as a prequisite for developing an acepatable
framework.  Hence this study recognizes that such a framework must show clear logical
relationships among the various indicators and dimensions of resilience and provide logical
linkages between their abstraction and empirical requirements. The geometric based mathematical
modeling approach we have adopted shows these relationships and provides the linkage between
conceptual model and operational requirements.  Based on this, mathematical functions were
developed to establish logical relationships among key socio-technical parameters and quantities
that characterize the community resilience system, thus infusing a theoretical basis into the
framework. To enhance the integration of both technical and non-technical communal resiliency
factors and reduce model complexity, the conceptual framework was defined using a minimum
number of integrated components and interactions. This approach allows the adoption of a soft
computing tool for model analysis.  While the study developed a template for data collection and

illustrated its application, the template still relies on subjective opinions of experts which may be seen as a drawback of the model. Hence further research is suggested to explore the automation and standardization of the R-FIS input process by integrating with web based socio-economic and ecological rankings or indices of communities. Yet, from computational and operational perspectives, the adoption of a fuzzy inference system as an analytical tool is presented as a viable approach for harnessing the opinions and experiences of experts and residents.

In conclusion, this study which is centered on the need for an acceptable template to measure flood resilience examines the challenges, conceptual constraints and construct ramifications that have complicated the development of an operational framework for measuring the resilience of communities prone to flood hazard. Although the proliferation of conceptual models and frameworks for understanding resilience has indeed posed some challenges for development of an acceptable scenario-based measurement framework, there has been evidence of rich multidisciplinary insights resulting from the continuously evolving collaborative platforms for driving resilience research, policy and discourse. Non-linearity, multiple feedbacks and other sources of complexity constitute major challenges to achieving operational practicality and model tractability while maintaining reasonable validity. There has also been the challenge of compatibility between the natural and human variables due to the well recognized complexity inherent in community resilience. In terms of insights, the models from this study provide some explanations into the relationships existing among resilience factors and dimensions. For instance, the importance of good community governance, processes and resource utilization systems becomes obvious in the various scenario analyses. Furthermore, the model was able to document the relative impact of variables that contribute to or detract from resilience. Although only sample values were used, the model application was able to illustrate the relative impacts that varying levels of institutional strength and resource availability, for example, have on progress toward resilience at a place.

Hence, the R-FIS provides a pathway for dealing with challenges of data issues such as missing data, spatiotemporal variations, and the use of subjective information because the critical input variables are locally and/or contextually defined. Thus, the proposed framework offers a viable approach for measuring flood resilience even when there are limitations of data availability and compatibility.

522

**Acknowledgements**

*This work is part of a research carried out under the Fulbright African Research Scholar Program Award (2017/18) funded by the United States Government.*

**6.0 References**

Adnan, M. M., Sarkheyli, A., Zain, A. M.,and Haron, H.  Fuzzy logic for modeling machining process: a review. *Artificial Intelligence Review , 43* (3), 345-379, https://doi.org/10.1007/s10462-012-9381-8, 2015.

Ashraf, M.,and Routray, K. K. Perception and understanding of drought and coping strategies of farming house holds in north-west Balochistan. *International Journal of Disaster Risk Reduction , 5*, 49-60, https://doi.org/10.1016/j.ijdrr.2013.05.002, 2013.

Begg, C., Walker., G.,and Kuhlicke, C.  Localism and flood risk management in England:the creation of new inequalities? *Government and Policy , 33* (4), 685-702, https://doi.org/10.1068/c12216, 2015.

Brown, E. D.,and Williams, B. K.  Resilience and Resource Management. *Environmental Management , 56(6)*, 1416–1427, https://doi.org/10.1007/s00267-015-0582-1, 2015.

Cai, H., Lam, N. S., Zou, L.,and Qiang, Y. Modeling the Dynamics of Community Resilience to Coastal Hazards Using a Bayesian Network. *Annals of the American Association of Geographers , 108(5), 1260-1279,  https://doi.org/10.1080/24694452.2017.1421896, 2018.

Cohen, O., Bolotin, A., Lahad, M., Goldberg, A.,and Aharonson-Daniel, L.  Increasing sensitivity of results by using quantile regression analysis for exploring community resilience. *Ecological Indicators , 66*, 497-502, https://doi.org/10.1016/j.ecolind.2016.02.012 , 2016.

Cohen, O., Goldberg, A., Lahad, M.,and Aharonson-Daniel, L. Building resilience: The relationship between information provided by municipal authorities during emergency situations and community resilience. *Technological Forecasting and Social Change , 121*, 119-125, https://doi.org/10.1016/j.techfore.2016.11.008 , 2017.

Costache, A.  Conceptual delimitations between resilience, vulnerability and adaptive capacity to extreme events and global change. *Annals of Valahia University of Targoviste. Geographical Series , 17* (2), 198-205, https://doi.org/10.1515/avutgs-2017-0018 , 2107.

Cutter, S. L.  Linkages between Vulnerabilty and Resilience. In S. Fuchs,and T. Thaler, *Vulnerabilty and Resilience to Natural Hazards* (pp. 257-270). New York: Cambridge Press, 2018

Cutter, S. L., Ash, K. D.,and Emrich, C. T.  Urban–Rural Differences in Disaster Resilience. *Annals of the*
*American Association of Geographers , 106* (6), 1236-1252,
https://doi.org/10.1080/24694452.2016.1194740, 2016
Cutter, S. L., Barnes, L., Berry, M.,and Burton, C. A place-based model for understanding community
resilience to natural disasters. *Global environmental change , 18* (4), 598-606,
https://doi.org/10.1016/j.gloenvcha.2008.07.013  2008.
Duy, P. N., Chapman, L., Tight, M., Thuong, L. V., Ph.D,and Linh, P. N.  Urban Resilience to Floods in
Coastal Cities: Challenges and Opportunities for Ho Chi Minh City and Other Emerging Cities in Southeast
Asia. *Journal of Urban Planning and Development , 144* (1), 2018.
Esteban, M., Tsimopoulou, V., Mikami, T., Yun, N. Y., Suppasri, A.,and Shibayama, T.  Recent tsunamis
events and preparedness: Development of tsunami awareness in Indonesia, Chile and Japan.
*International Journal of Disaster Risk Reduction , 5*, 84-97, https://doi.org/10.1016/j.ijdrr.2013.07.002 ,
565    2013.

Fekete, A.,and Montz, B.  Vulnerability: an introduction. In S. Fuchs,and T. Thaler, *Vulnerability and*
*resilience to natural hazards.* Cambridge University Press, 2018.
Filion, P.,and Sands, G. Enhancing Hazard Resilience among Impoverished Urban Communities in Ghana:
The Role of Women as Catalysts for Improvement. In P. Filion,and G. Sands, *In Cities at Risk: Planning for*
*and Recovering from Natural Disasters* (pp. 31-46). Routledge , 2016.
Fisher, L.  Disaster responses: More than 70 ways to show resilience. *Nature , 518* (7537), 35-35,
https://doi.org/10.1038/518035a, 2015.
Folke, C.  Resilience:theemergenceofaperspectivefor social–ecologicalsystemsanalyses.
*GlobalEnvironmental Change , 16*, 253–267,  https://doi.org/10.1016/j.gloenvcha.2006.04.002 , 2006.
Folke, C., Carpenter, S., Elmqvist, ,. T., Gunderson, L., Holling, C. S.,and Walker, B. Resilience and
sustainable development: building adaptive capacity in a world of transformations. *AMBIO: A journal of*
*the human environment , 31* (5), 437-440,  https://doi.org/10.1579/0044-7447-31.5.437, 2002
Fuchs, S.,and Thaler, T.  *Vulnearabilty and Resilience to Natural Hazards.* (S. Fuchs,and T. Thaler, Eds.)
New York: Cambridge Press , 2018.
Guo, L., He, B., Chang, M., Chang, Q., Li, Q., Zhang, K., et al.  A comprehensive flash flood defense system
in China: overview, achievements, and outlook. *Nat Hazards* , 1-14 , https://doi.org/10.1007/s11069-
582    018-3221-3, 2018.

Hammond, M. J., Chen, A. S., Djordjević, S., Butler, D.,and Mark, O. Urban flood impact assessment: A
state-of-the-art review. *Urban Water Journal , 12* (1), 14-29 ,
https://doi.org/10.1080/1573062X.2013.857421, 2015.
Ibanez, G. E., Buck, C. A., Khatchikian, N.,and Norris, F. H.  Qualitative Analysis of Coping Strategies
Among Mexican Disaster Survivors. *Anxiety, Stress, and Coping , 17* (1), 69-85 ,
https://doi.org/10.1080/10615800310001639628, 2004.
Jha, A. K., Bloch, R.,and Lamond, J. *Cities and Flooding: A guide to Integrated Urban Flood Risk*
*Management the 21st Centuries .* Washington : The World bank , 2012.
Joseph, R., Proverbs, D.,and Lamond, J.  Resilient reinstatement: what can we learn from the 2007
flooding in England? In D. Proverbs,and C. A. Brebbia, *Flood Recovery, Innovation and Response IV ,* v
184, 175-186, . WIT Press , 2014.
Keating, A., Campbell, K., Szoenyi, M., McQuistan, C., Nash, D.,and Burer, M.  Development and testing
of a community flood resilience measurement tool. *Natural Hazards and Earth System Sciences. 17(1),*
77-101,  https://doi.org/10.5194/nhess-17-77-2017, 2017.
Kron, W.  Flood Risk = Hazard • Values • Vulnerability. *Water International , 30* (1), 58-68,
https://doi.org/10.1080/02508060508691837, 2005.
Lee, A. V., Vargo, J.,and Seville, E.  Developing a Tool to Measure and Compare Organizations' Resilience.
*Natural Hazards Review , 14* (1), 29-41 , **https://doi.org/10.1061/(ASCE)NH.1527-6996.0000075**, 2013.
Lincy, G. R.,and John, C. J.  A multiple fuzzy inference systems framework for daily stock trading with
application to NASDAQ stock exchange. *Expert Systems with Applications: An International Journal , 44*
(C), 13-21 , https://doi.org/10.1016/j.eswa.2015.08.045, 2016.
Mallakpour, I.,and Villarini, G.  The changing nature of flooding across the central United States. *Nature*
*Climate Change , 5*, 250–254, https://doi.org/10.1038/nclimate2516, 2015.
Mamdani, E. H.,and Assilian, S.  An experiment in linguistic synthesis with a fuzzy logic controller.
*International Journal of Man-Machine Studies , 7* (1), 1-13 , https://doi.org/10.1016/S0020-
608    7373(75)80002-2, 1975.

Mavhura, E., Manyena, S. B., Collins, A. E.,and Manatsa, D.  Indigenous knowledge,coping strategies and
resilience to floods in Muzarabani,Zimbabwe. *International Journal of Disaster Risk Reduction , 5*, 38-48,
https://doi.org/10.1016/j.ijdrr.2013.07.001, 2013.
Mogollón, B., Frimpong, E. A., Hoegh, A. B., and Angermeier, P. L.  Recent changes in stream flashiness
and flooding, and effects of flood management in North Carolina and Virginia. *JAWRA Journal of the*
*American Water Resources Association , 52* (3), 561-577, **https://doi.org/10.1111/1752-1688.12408**,
615    2016.


Montz, B.  Emerging Issues and Challenges: Natural Hazards. *Journal of Contemporary Water Research*
*and Education , 142*, 42-45, https://doi.org/10.1111/j.1936-704X.2009.00051.x , 2009.
Multi-dimensional hurricane resilience assessment of electric power systems. *Structural Safety , 48*, 15-
24 , https://doi.org/10.1016/j.strusafe.2014.01.001 , 2014.
Norris, F. H., Stevens, S. P., Pfefferbaum, B., Wyche, K. F.,and Pfefferbaum, R. L. Community resilience as
a metaphor, theory, set of capacities and strategy for disaster readiness. *Community Psychology, 4 (1-2)*,
127-50, https://doi.org/10.1007/s10464-007-9156-6, 2008.
North Carolina. (2019). *NC Flood Mapping Program*. Retrieved May 04, 2019, from Flood. NC.Gov :
https://flood.nc.gov/ncflood/

NRC. *Disaster resiience - a national imperitive.* Washington, DC: National Academies press,  2012.
Oladokun, V. O.,and Emmanuel, C. G. Urban Market Fire Disasters Management in Nigeria: A Damage
Minimization based Fuzzy Logic Model Approach. *International Journal of Computer Applications , 106*
630    (17) , 2014.

Oladokun, V. O., Proverbs, D. G.,and Lammond, J.  Measuring flood resilience: A fuzzy logic approach.
*International Journal of Building Pathology and Adaptation* , 35(5), 470-487,
https://doi.org/10.1108/IJBPA-12-2016-0029, 2017.
Park, J., Seager, T. P., Rao, P. S., Convertino, M.,and Linkov, I.  Integrating risk and resilience approaches
to catastrophe management in engineering systems. *Risk Analysis , 33* (3), 356-367 ,
https://doi.org/10.1111/j.1539-6924.2012.01885.x , 2013.
Parsons, M., Glavac, S., Hastings, P., Marshall, G., McGregor, J., McNeill, J., Morley, P., Reeve, I.,and
Stayner, R. Top-down assessment of disaster resilience: Aconceptual framework using coping and
adaptive capacities. *International Journal of Disaster Risk Reduction , 19*, 1-11 ,
https://doi.org/10.1016/j.ijdrr.2016.07.005, 2016.
Pitt County Development Commision. (2019). *Geography and Climate*. Retrieved May 04, 2019, from Pitt
County Development Commision: http://locateincarolina.com/geography-climate/
Qiang, Y.,and Lam, N. S. The impact of Hurricane Katrina on urban growth in Louisiana: An analysis using
data mining and simulation approaches. *International Journal of Geographical Information Science , 30*
(9), 1832–52, https://doi.org/10.1080/13658816.2016.1144886, 2016.
Rose, A.  Broader Dimensions of Economic Resilience. In *Defining and Measuring Economic Resilience*
*from a Societal, Environmental and Security Perspective. Integrated Disaster Risk Management.*
Singapore: Springer , 2017.
Schelfaut, K., Pannemans, B., van der Craats, I., Krywkow, J., Mysiak, J.,and Cools, J.  Bringing
floodresilienceintopractice:theFREEMAN project. *Environmental Science and Policy , 14* (7), 825-833 ,
https://doi.org/10.1016/j.envsci.2011.02.009 , 2011.
Scoones, I. *Sustainable rural livelihoods: a framework for analysis.* Institute for Development Studies,
Brighton, UK, http://opendocs.ids.ac.uk/opendocs/handle/123456789/3390, 1998.
Shah, A. A., Ye, J., Abid, M., Khan, J., and Amir, S. M. Flood hazards: household vulnerability and
resilience in disaster-prone districts of Khyber Pakhtunkhwa province, Pakistan. *Natural Hazards , 93*
(1), 147-165, https://doi.org/10.1007/s11069-018-3293-0, 2018.
Sharifi, A. A critical review of selected tools for assessing community resilience. *Ecological Indicators ,*
*69*, 629–647, https://doi.org/10.1016/j.ecolind.2016.05.023 , 2016,
Su, Y. S. Discourse, Strategy, and Practice of Urban Resilience against Flooding. *Business and*
*Management Studies , 2* (1), 73-87, https://doi.org/10.11114/bms.v2i1.1348 , 2016.
Su, Y. S. Urban Flood Resilience in New York City, London, Randstad, Tokyo, Shanghai, and Taipei.
*Journal of Management and Sustainability , 6* (1), 92, https://doi.org/10.5539/jms.v6n1, 2016.
Swalheim., S.,and Dodman, D. . *Building resilience: how the urban poor can drive climate adaptation.*
London: IIED, 2008.
Szoenyi, M.,and Nash, D. *Measuring flood resilience – our approach.* Zurich: Zurich Insurance Company
Ltd, 2016.
Thomas, J. A.,and Mora, K. Community resilience, latent resources and resource scarcity after an
earthquake: Is society really three meals away from anarchy? *Natural hazards , 74* (2), 477-490,
https://doi.org/10.1007/s11069-014-1187-3, 2014.
Tompkins, E.,and Adger, W. N. Does adaptive management of natural resources enhance resilience to
climate change? *Ecology and society , 9* (2), https://www.jstor.org/stable/26267677, 2004.
Trogrlić, R. Š., Wright, G. B., Adeloye, A. J., Duncan, M. J.,and Mwale, F. Taking stock of community-
based flood risk management in Malawi: different stakeholders, different perspectives. *Environmental*
*Hazards , 17* (2), https://doi.org/10.1080/17477891.2017.1381582, 2018.
Wing, O. E., Bates, P. D., Smith, A. M., Sampson, C. C., Johnson, K. A., Fargione, J., and Morefield, P.
Estimates of present and future flood risk in the conterminous United States. *Environmental Research*
*Letters , 13*, https://doi.org/10.1088/1748-9326/aaac65, 2018.
Zadeh, L. A. Fuzzy logic= computing with words. *Fuzzy Systems, IEEE Transactions on , 4* (2), 103-111,
679 2016.

Zou, L., Lam, N. S., Cai, H.,and Qiang, Y. Mining Twitter Data for Improved Understanding of Disaster
Resilience. *Annals of the American Association of Geographers , 1*–20,
https://doi.org/10.1080/24694452.2017.1421897 , 2016,

**Table 1 .** Resilience dimensions and descriptions of input factors influencing their states

| Resilience Dimensions | Resilience input factors |
|---|---|
| **1.** Hazard Absorbing capacity **H** | 1. Level of infrastructure in terms of sophistication and adequacy. Effectiveness of FRM measures such as flood and shoreline defenses, forecast and warning system, <br> 2. Redundant capacities. Evidence of alternatives in critical utilities, evacuation routes, communication and energy infrastructures, hospitals, police posts, supermarkets. <br> 3. Evidence of redundant housing capacity. <br> 4. Ecological defenses and buffer. Evidence of complementary use of nature to improve threshold, e.g. using landscaping and topography, natural drainage and canals, vegetation cover, rain/storm water harvesting, permeable pavements, etc. <br> 5. Residents coping capacity. Evidence of large portion of populace with previous flood experience, awareness, cohesion and place attachment <br> 6. Evidence of stable or growing population in spite of past events. <br> 7. Educational and literary level of populace <br> 8. Evidence of social and communal clusters to enhance coping through support, meaning, avoidance etc., e.g. church, local sport team, ethnic clusters. <br> 9. Presence of critical and strategic institutions of national importance, e.g. university, military base, major ports, etc. <br> 10. Evidence of technology driven information dissemination, e.g. social media, sms (Ashraf and Routray, 2013; Cohen et al., 2017; Esteban et al., 2013; Ibanez et al., 2004; Lee et al., 2013; Mavhura et al., 2013) |
| **2.** Resource Availability **G** | 1. Evidence of budgetary provision for, or commitment to, flood risk management. <br> 2. Evidence of thriving economic activities in the community, e.g. size of local GDP <br> 3. Evidence of economic strength of residents, e.g. per capita income, income level, housing value, savings, cooperative societies, etc. <br> 4. Evidence of political, institutional and economic influence that can attract grants and funds from national or regional sources, e.g. population <br> 5. Evidence of adoption of flood insurance plans. <br> 6. Availability of land for relocation development beyond or outside the flood plains. <br> 7. Evidence of community capital and community natural assets accessible for reconstruction, e.g. forest resources, granite and quarry deposits. <br> 8. Economic status of the 'parent' entity, e.g. the state's or country's GDP (Filion and Sands, 2016; Rose, 2017; Swalheim and Dodman, 2008; Thomas and Mora, 2014) |
| **3.** Community Processes and Resource Utilization **θ** | 1. Evidence of good governance <br> 2. Level of ease of doing business <br> 3. Evidence of strong institutions such as judiciary, police, media, and public service <br> 4. Evidence of culture of law and order. <br> 5. Ranking of internationally recognized bodies like Transparency International, World Bank, UN, CIA, etc. on the above (Begg et al., 2015; Brown and Williams, 2015; Cohen et al., 2016; Rose, 2017; Tompkins et al., 2004) |

**Table 2.** Fuzzy inference linguistic variables term set and membership functions (Adnan et al., 2015; Oladokun and Emmanuel, 2014)

| Linguistic Variables | Term sets | Membership function |
|---|---|---|
| Hazard Absorbing | Low | PiMfunction |
| Capacity H | High | GbellMf |
| **Input 1** | Very High | SMfunction |
| Resource | Very Low | ZMfunction |
| Availability G. | Low | GaussianMfunction |
| **Input 2** | High | SigMfunction |
| Resource Utilization | Poor | PiMfunction |
| Processes $\theta$. | Good | GaussianMfunction |
| **Input 3** | Excellent | PiMfunction |
| | Very Low | Zmfunction |
| Resilience $R_i$ | Low | Gauss2Mfunction |
| **Output** | Moderate | GbellMfunction |
| | High | PiMfunction |
| | Very High | PiMfunction |

**Table 3 Sample rules of the R-FIS 27 Rule Base** (Rules and weights to be determined by experts and/or stakeholders)

| Rules premise | Rules Consequence | Weight |
|---|---|---|
| If (**H** is Low) & (**G** is Very Low ) & ($\theta$ is Poor) THEN | (Resilience is very low) | 1 |
| If (**H** is Low) & (**G** is Low) & ($\theta$ is Excellent ) THEN | (Resilience is Low) | 0.8 |
| If (**H** is Low) & (**G** is High) & ($\theta$ is Excellent) THEN | (Resilience is Moderate) | 0.8 |
| If (**H** is High) & (**G** is High) & ($\theta$ is Excellent) THEN | (Resilience is Moderate) | 1 |
| If (**H** is Very High) & (**G** is Very  Low) & ($\theta$ is Good) THEN | (Resilience is High) | 0.7 |
| If (**H** is Very High) & (**G** is High) & ($\theta$ is Good) THEN | (Resilience is High) | 1 |
| If (**H** is Very High) & (**G** is High) & ($\theta$ is Excellent ) THEN | (Resilience is Very High) | 1 |

**Table 4. Linguistic variables input template** (to be used with Table 1 as a scoring guide)

| Linguistic Variables Dimension | Tick the grey box next to your linguistic rating | Tick the grey box that best reflects your score of your linguistic rating | | | |
|---|---|---|---|---|---|
| Hazard Absorbing Capacity **(H)** | Low | | 1 | 2 | 3 |
| | Moderate | | 4 | 5 | 6 |
| | High | | 7 | 8 | |
| | Very High | | 9 | 10 | |
| Resource Availability **(G)** | Low | | 1 | 2 | 3 |
| | Moderate | | 4 | 5 | 6 |
| | High | | 7 | 8 | |
| | Very High | | 9 | 10 | |
| Resource Utilization Processes **(θ)** | Poor | | 1 | 2 | 3 |
| | Good | | 4 | 5 | 6 |
| | Very Good | | 7 | 8 | |
| | Excellent | | 9 | 10 | |

Location/city
Date of assessment
Assessors' name

**Table 5 Study locations- demographic and topographic summary** (Source:  http://census.gov and United States Geological Survey Topographic Maps)

| | Windsor NC | Greenville  NC | Norfolk  VA |
|---|---|---|---|
| Location type | Small town | City | Large city |
| Types flood | River/storm/ rain | River /storm/ Rain | Coastal /river rain/storm |
| Total Population* | 3,630 | 84,554 | 242,803 |
| Male * (%) | 59.3 | 45.8 | 51.8 |
| Female* (%) | 40.7 | 54.2 | 48.2 |
| Median income * ($) | 29,063 | 34,435 | 44,480 |
| Poverty rate * (%) | 27.8 | 32.5 | 21 |
| Median Age*  (yr) | 38.6 | 26.0 | 29.7 |
| Under 14* (%) | 12.4 | 15.9 | 17.7 |
| 75 above* (%) | 8.7 | 4.3 | 4.6 |
| US Citizenship *(%) | 97.9 | 96.8 | 96.6 |
| Non English speaking *(%) | 5.83 | 6.74 | 10.3 |
| No of Households* | 1,088 | 36,071 | 85,485 |
| Family household* (%) | 61.2 | 46.3 | 58.7 |
| Average household size* | 2.29 | 2.18 | 2.43 |
| Household with individuals above 65* (%) | 34.1 | 14 | 20.3 |
| No of Housing units* | 1,193 | 40,564 | 95,018 |
| housing units occupied* (%) | 91.2 | 88.9 | 91.0 |
| Mean property Value ($)* | 93,800 | 147,100 | 193,400 |
| ** Elevation  (meter ) | 7.62 | 17.07 | 9.14 |

**Table 6. Input scoring and R-FIS resilience index output**

| Experts Scoring / Community | Model Input | | | | | | Model Output |
|---|---|---|---|---|---|---|---|
| | Hazard Absorbing Capacity (**H**) | | Resource Availability (**G**) | | Resource Utilization Processes (**θ**) | | Resilience Index **R** |
| | Linguistic Score | Score | Linguistic Score | Score | Linguistic Score | Score | |
| Norfolk, VA | High | 7.0 | High | 8.0 | Good | 6.0 | **0.836** |
| Greenville, NC | High | 8.0 | Moderate | 6.0 | Very Good | 8.0 | **0.9** |
| Windsor, NC | Moderate | 4.0 | Low | 2.0 | Very Good | 8.0 | **0.477** |

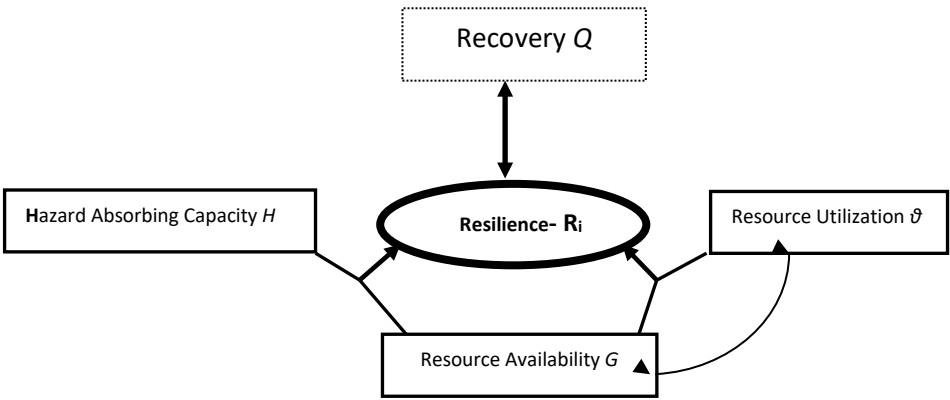

**Figure 1.** Resilience measuring conceptual framework

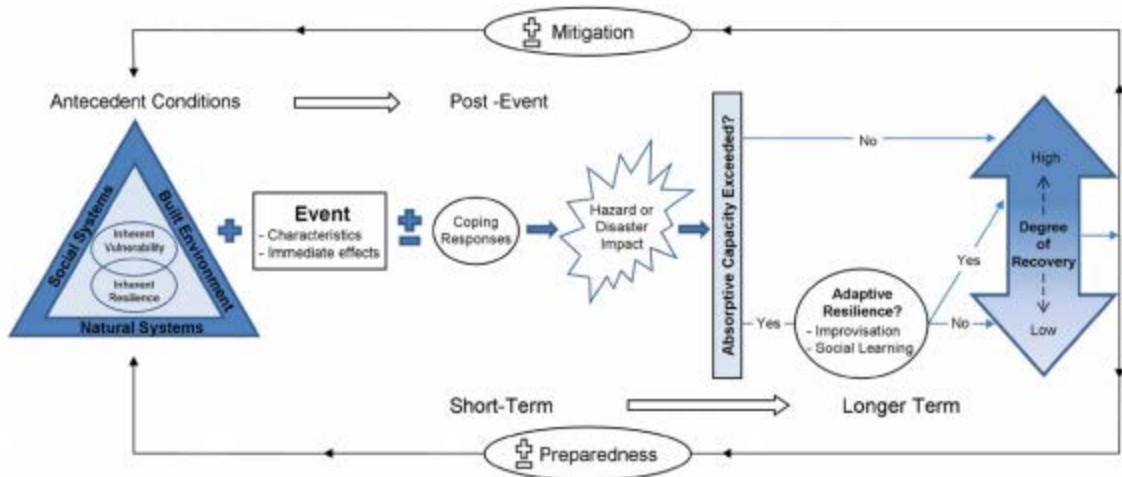

Schematic representation of the disaster resilience of place (DROP) model.

**Figure 2**. The Disaster Resilience of Place (DROP) model reproduced from Cutter et al, (2008). A place-based model for understanding community resilience to natural disasters. This model illustrates the interelationship between resilience and recovery within the hazard–resilience system.

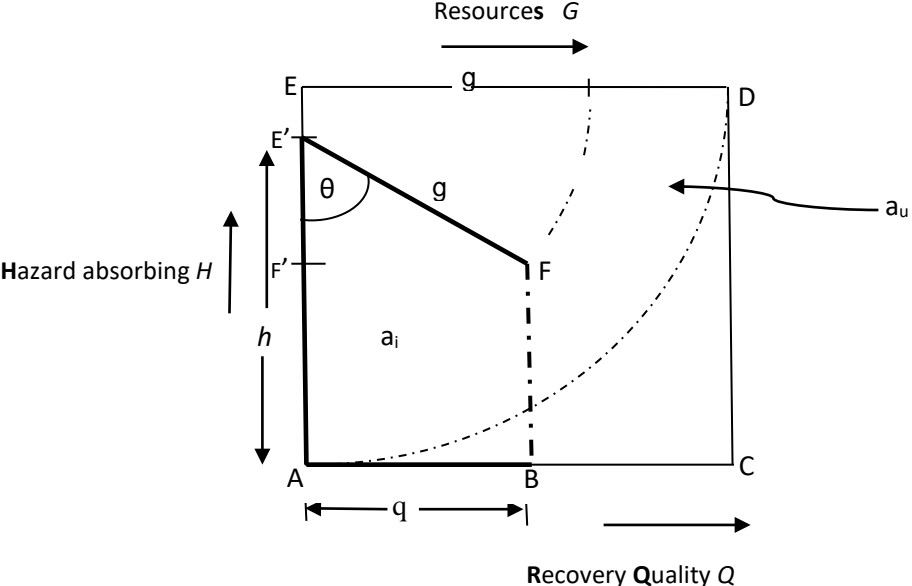

**Figure 3:** Resilience conceptual model. A geometric model used to derive appropriate mathematical relationships of the proposed framework and provide some insights

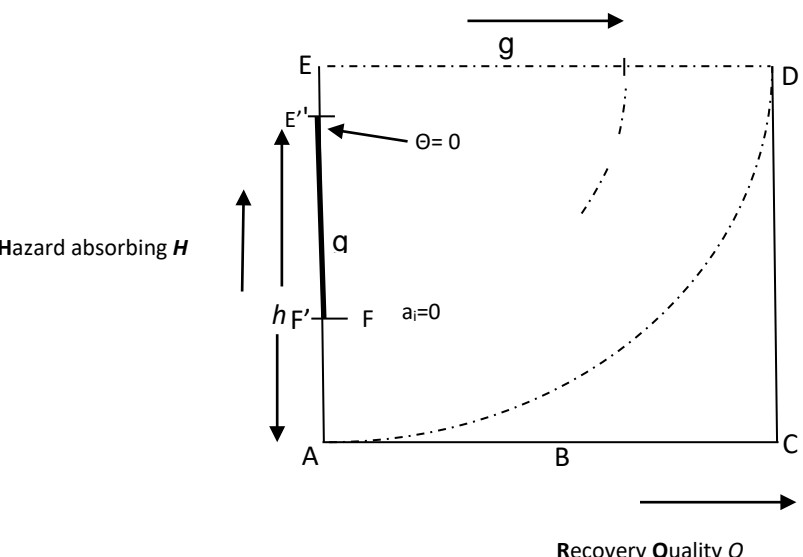

**Figure 4.** Resilience area = 0 when ρ= Sin Θ= 0. A variation of model Figure 3 depicting an extreme case of a community with zero efficiency in resource utilization.

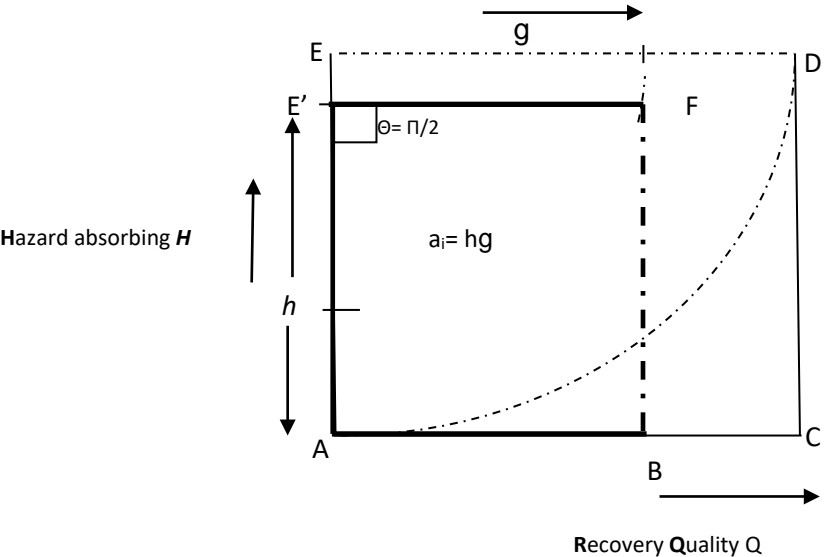

**Figure 5.** Resilience area ($a_i$ = hg). A variation of model Figure 3 depicting an extreme case of a community with a perfect resource utilization system (efficiency of 1.0) which maximizes recovery resources' g on absorbing capacity h.

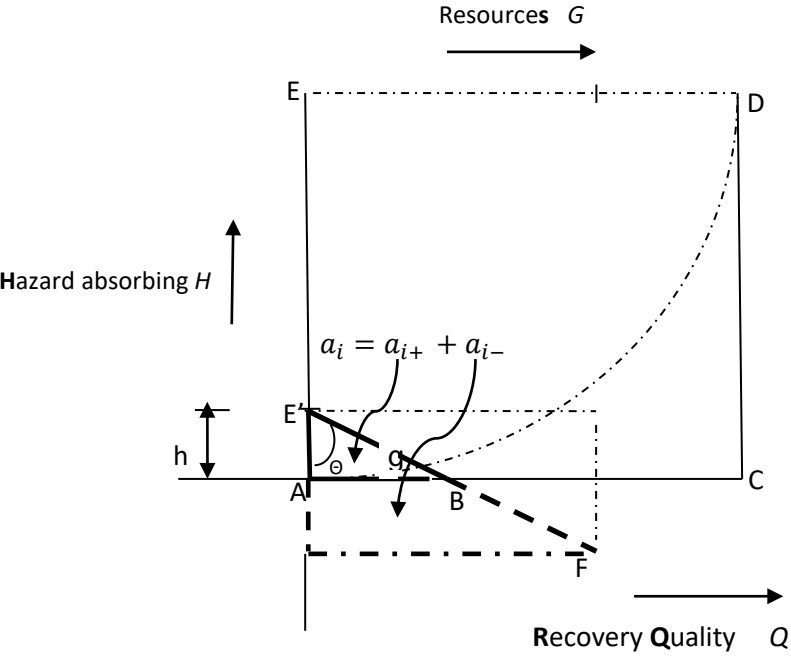

**Figure 6.** Resilience as absorbing capacity approaches zero

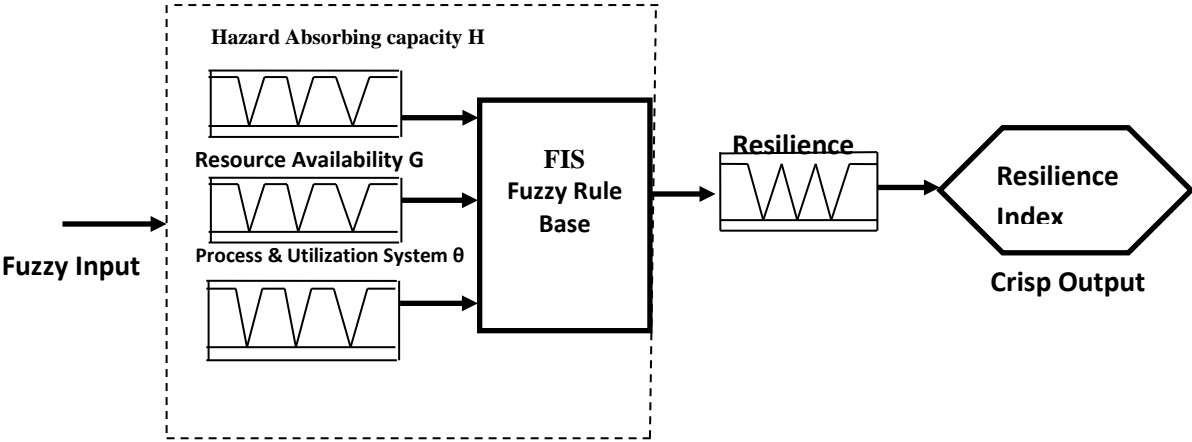

**Figure 7.** Resilience fuzzy inference systems

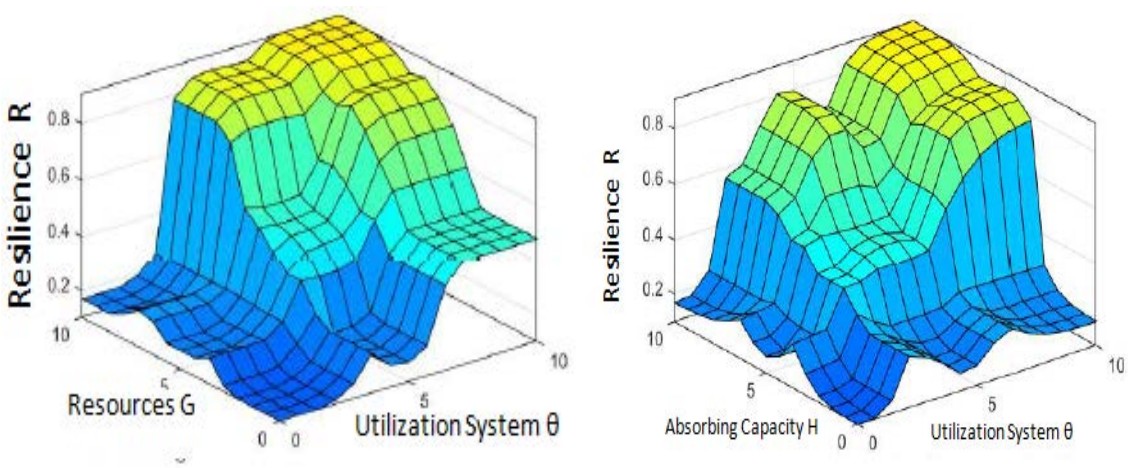

**Figure 8.** Examples of resilience output surface plots.

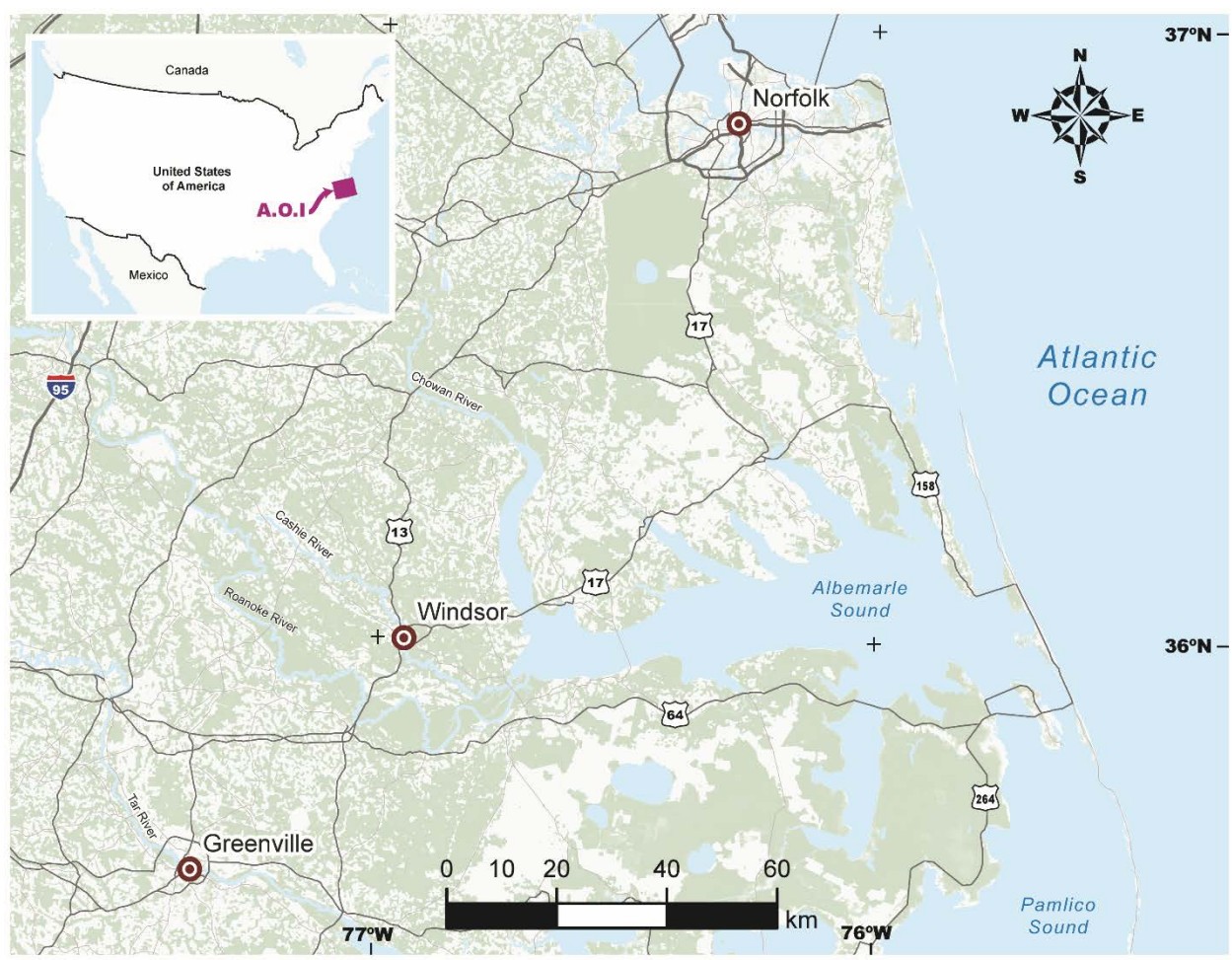

**Figure 9.** The study area on map showing Greenville, NC; Windsor, NC and Norfolk VA
Source: Produced in the GIScience Center, East Carolina University

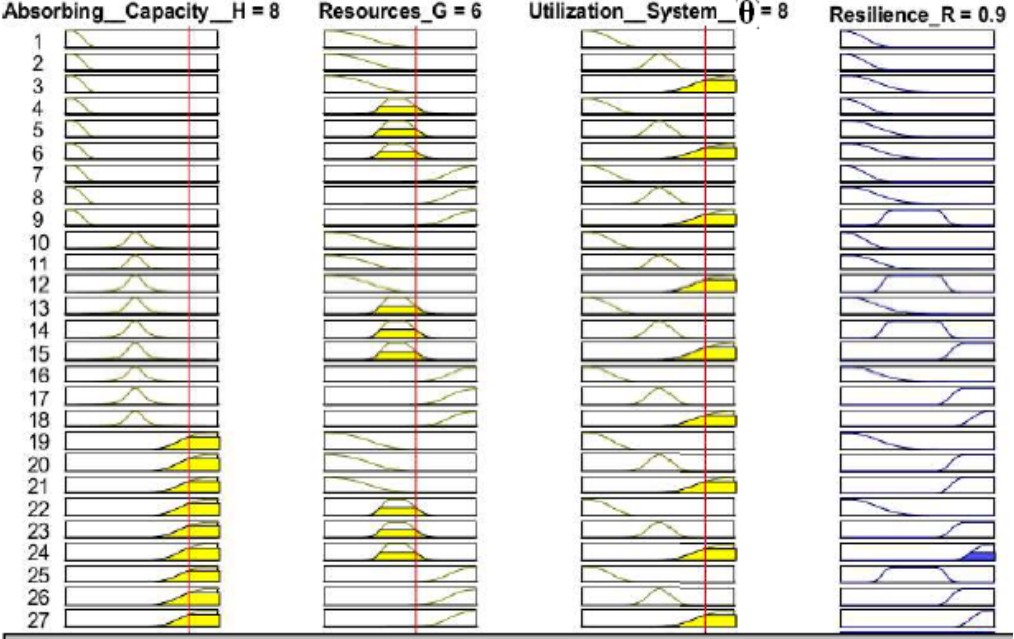

**Figure 10.** Rule setting and output for Greenville
🟨 Active input membership functions 🟦 Active output membership function