# Peer review of "Towards Measuring Resilience of Flood Prone Communities: A Conceptual Framework V.O.Oladokun1 and B.E.Montz2 2 1Department of Industrial and Production Engineering, University of Ibadan, Ibadan, Nigeria 3 2Department of Geography, Planning, & Environment, East Carolina Univer"

_Natural Hazards and Earth System Sciences, 2018_

## Referee Comment (RC1) · Anonymous Referee #1 · 3 Sep 2018

This paper adds to expanding literature on disaster resilience measurement. The primary purpose of the study is to develop a mathematical model based on the U.S. National Academies definition of resilience ("the ability to prepare and plan for, absorb, recover from, and more successfully adapt to adverse events") and then implement the model for three flood-prone communities using a fuzzy logic equivalent. The background on the development of operational resilience measurement models is good, although it does rely primarily on relatively few papers (e.g. Cai et al. 2018; Cutter 2018; Keating et al. 2017; Zou et al. 2018), and perhaps misses other community resilience measurement efforts such as Zurich's Flood Resilience program. I am also concerned that the definitional discourse does not adequately describe the complexities and variability in the meaning of resilience—is it applied to a particular system, event, or more broadly to capture community abilities as the NRC definition is designed to do? I would encourage the authors to reduce the definitional discussion and simply select and then justify the definition they prefer to use (e.g. NRC 2012) as the basis for their conceptual model. In the formulation of the conceptual mode (Figure l,) the authors assume that resilience leads to recovery (the outcome of interest). How does the conceptual model line up with their preferred definition? While an attempt was made on p. 5-6 to do this, most of the discussion is focused on recovery or the recovery spectrum. So, how can the operationalization of a definition that includes recovery also be used to measure an outcome, also labeled recovery? The authors need to clearly distinguish resilience (an outcome in and of itself) from recovery or at a minimum more clearly articulate they are describing resilience-type capacities within communities that influence flood recovery. It seems to me that the conceptual model is oriented to flood recovery (p. 7) rather than resilience per se. Later on in the paper, they use the resilience index as the output (Table 6), but this is not found in the conceptual model as described in Figure 1). What is unique about the context of flood hazards in the model, or could it equally apply to any natural hazard impact in a community? The bulk of the paper describes the mathematics of the model and its implementation, but again I wonder as to whether the model is describing and/or modeling resilience. What is the source of the resilience input factors? Were the inputs verified to see if the model worked? In the "hypothetical" analysis who determined the inputs (e.g. who did the assessment as to the values of the inputs)? There is no explanation of this in Section 4 Model Application, just a very generic text about the study location. When the "results" appear, they are more like a description of the tool and how it can be used rather than results based on empirical and/or qualitative assessments. Thus, the information presented in the manuscript does not support the results as presented. In addition, the discussion and conclusion section is not especially robust either and in many ways rehashes the literature review rather than presenting new and innovative findings related to resilience in flood prone communities. This paper could be significantly improved by re-framing it as a methodological contribution where the conceptual model and its mathematical expression is more fully articulated including all the requisite input variables including the sources. Then the fuzzy logic scoring template/tool can be described in more detail. In order to test the model, however, the authors would need to generate at least a small sample of stakeholders to complete the input variable assessments as a measure of the validity of the effort. This is a difficult paper to assess given how much of it seems focused on the modeling (Figures 2-6) and recovery quality, yet in these same figures there's no mention of the other two components (resource availability and resource utilization processes) unless these are both subsumed under resources per Figure 3. As a reader I do not understand the model and its conversion to a type of resilience index (the stated output). Whether this is a function of my lack of familiarity with mathematical modeling as used here or the authors' explanation of it is uncertain. Either way, the manuscript needs a rewrite to make it appeal more directly to the journal's readership.

---

## Referee Comment (RC2) · Anonymous Referee #2 · 25 Oct 2018

This manuscript contributes to broad research field on community resilience and aims to develop framework on measuring resilience. After examining and discussing the challenges of different definition and concepts in this context, the authors presented a conceptual and mathematical model as well as applied a fuzzy logic approach to generate a resilience index, which was applied in three flood-prone communities in the US (North Carolina and Virginia).

The manuscript is in general well written but the structure and the different level of information provided in the sections challenge the reader to follow the argumentation of authors and relate the different parts of the framework. For example, the introduction

provides a selected overview on the topic and challenges of community resilience and different frameworks to measure resilience. However, the focus is on different definition of resilience and not on the differences in approaches to measure resilience, which are only mentioned but not explain. I see here a high potential to reduce the definition discussion and provide more details on measuring resilience. Moreover, I suggest to present also a clear objective for the study, which would help to follow the structure of the manuscript. Perhaps a flow chart showing the interrelation of the different models would also increase the understanding of the chosen structure. Furthermore, in the design of the model (also including Fig. 1) it is not very well explained why resilience leads than to recovery, as one part of resilience would be 'how the community is able to recover?' and thus this parameter should contribute to measure resilience.

The authors illustrated in section 2.5 some extreme cases and showed the gained insights to model structure, however it also shows the limitation of the model regarding dynamical change, e.g. if you consider in case 1 that you have no efficiency in the resource utilization processes = no resilience, then your model ignore any preparedness and coping capacity. I would agree on a long term but you measuring only in static manner, thus should not there be a difference of communities with different hazard absorbing capacity? The authors are also encourage to provide more thoughts about their assumption of that 'negative' resilience is another expression of vulnerability.

The structure of section 3 is confusing because in the beginning it is not clear why the fuzzy logic is addressed and how it is related to the previous sections. Furthermore, from 3.2. onwards more detailed information on chosen criteria for selecting variables, number of rules, type of membership functions, weights ... are needed. Currently, in this section a lot of questions arise, but I see a high potential to improve the whole manuscript if you revise this section (see detailed remarks in the attached file).

In general the application in case study is only a very vague description and it is not clear on which assumption you based your hypothetical input score. I also would see an added value - if you stay with hypothetical inputs - to gain more insights on

the sensitive of your model with systematically testing of different input data but also on the rule setting and membership functions. The added-value would be a better understanding of model. The discussion and conclusion is very generic and needs to be rewritten regarding the points highlighted in the introduction, the (missing) objectives and the gained insights. I indicated different ways how the authors may restructure and rewrite the manuscript to show the added value of this study to the readers and scientific community. See detailed comments in the attached file.

Please also note the supplement to this comment:
https://www.nat-hazards-earth-syst-sci-discuss.net/nhess-2018-217/nhess-2018-217-RC2-supplement.pdf

**Supplement:**

[revised manuscript text omitted]

15 Figure 1: Resilience measuring conceptual framework

---

## Author Comment (AC1) · 30 Nov 2018

Victor O. Oladokun and Burrell E. Montz

montzb@ecu.edu

Received and published: 30 November 2018

REVIEWER'S COMMENT –Section 1a This paper adds to expanding literature on disaster resilience measurement. The primary purpose of the study is to develop a mathematical model based on the U.S. National Academies definition of resilience ("the ability to prepare and plan for, absorb, recover from, and more successfully adapt to adverse events") and then implement the model for three flood-prone communities using a fuzzy logic equivalent. AUTHORS' RESPONSE to REVIEWER COMMENT –Section 1a Comments reflect the broad scope of the paper

REVIEWER's COMMENT -Section 1b The background on the development of operational resilience measurement models is good, although it does rely primarily on relatively few papers (e.g. Cai et al. 2018; Cutter 2018; Keating et al. 2017; Zou et al. 2018), and perhaps misses other community resilience measurement efforts such as Zurich's Flood Resilience program. AUTHORS' RESPONSE to REVIEWER COM-MENT –Section 1b While efforts will be made to include other relevant papers, the authors wish to state that the Zurich flood resilience program was considered through one of the papers. The Zurich resilience measurement work was the focus of Keating et al (2017) that we cited. Quoting from paragraph 2 on 2nd page of Keating et al (2017), "The primary purpose of this paper is to present the Zurich Alliance community flood resilience measurement framework and associated tool, developed by an alliance of NGOs, academic institutions, and the private sector."

REVIEWER'S COMMENT –Section 2a I am also concerned that the definitional discourse does not adequately describe the complexities and variability in the meaning of resilience as it is applied to a particular system, event, or more broadly to capture community abilities as the NRC definition is designed to do? AUTHORS' RESPONSE to REVIEWER COMMENT –Section 2a This definitional discourse recognizes the community as a complex coupled system. Some of the variability and complexity were highlighted in section 1. For instance, we noted that the concept of resilience involves the interactions of several entities each defined by some social, economic, natural, technical and environmental dimensions, each of which we described as characterized by dynamic and complex spatiotemporal interactions.

REVIEWER'S COMMENT –Section 2b I would encourage the authors to reduce the definitional discussion and simply select and then justify the definition they prefer to use (e.g. NRC 2012) as the basis for their conceptual model. AUTHORS' RESPONSE to REVIEWER COMMENT –Section 2a This suggestion is noted and will be explored in the revision

REVIEWER'S COMMENT –Section 2b In the formulation of the conceptual mode (Figure I,) the authors assume that resilience leads to recovery (the outcome of interest).

How does the conceptual model line up with their preferred definition? While an attempt was made on p. 5-6 to do this, most of the discussion is focused on recovery or the recovery spectrum. So, how can the operationalization of a definition that includes recovery also be used to measure an outcome, also labeled recovery? AUTHORS' RESPONSE to REVIEWER COMMENT -Section 2b We appreciate the need to improve the clarity of Figure 1, firstly by using a two-way arrow arc to depict the interaction between resilience and recovery and secondly by enhancing the explanation of the figure with respect to the proposed model. We say our model is our interpretation of the definition as well as our understanding of the interactions within resilience. Our schematic model for instance recognizes that resilience enhances recovery or that recovery is an outcome of resilience whereby when a community, as a coupled system, becomes more resilient, its capacity to experience post disaster recovery increases. In other words, recovery, in terms of the time taken to attain post disaster recovery and the degree of recovery attained, are influenced by the resilience. This understanding is supported by the DROP resilience model as illustrated in Cutter, Barnes, Berry, & Burton (2008).

DROP model reproduced from Cutter et al 2008 Č Thus, our model implicitly suggests that recovery (i.e., recovery time and quality) can be a substitute for resilience. This is reasonable because post disaster recovery is driven by inherent resilience factors some of which we further explained in table 3 of this paper.

REVIEWER'S COMMENT –Section 3 The authors need to clearly distinguish resilience (an outcome in and of itself) from recovery or at a minimum more clearly articulate they are describing resilience-type capacities within communities that influence flood recovery. It seems to me that the conceptual model is oriented to flood recovery (p. 7) rather than resilience per se. Later on in the paper, they use the resilience index as the output (Table 6), but this is not found in the conceptual model as described in Figure 1). AUTHORS' RESPONSE to REVIEWER COMMENT –Section 3 It should be noted that this reviewer's comment underpins our argument about how the absence

СЗ

of consensus on definition leads to divergent interpretations of the interactions among the components of the resilience system. According to Cutter, Barnes, Berry, & Burton (2008), multiple definitions of resilience exist within the literature, with no broadly accepted single definition. Our schematic model for instance recognizes that resilience enhances recovery or that recovery is an outcome of resilience whereby when a community, as a coupled system, becomes more resilient, its capacity to experience post disaster recovery increases: pre-disaster resilience affects recovery, but post-disaster recovery can also affect resilience.

REVIEWER'S COMMENT – Section 4 What is unique about the context of flood hazards in the model, or could it equally apply to any natural hazard impact in a community? AUTHORS' RESPONSE to REVIEWER COMMENT – Section 4 The focus of the model is flood hazards. Flood hazards share characteristics with other natural hazards. This focus was also reiterated in the conclusion of the paper

REVIEWER'S COMMENT –Section 5 The bulk of the paper describes the mathematics of the model and its implementation, but again I wonder as to whether the model is describing and/or modeling resilience. AUTHORS' RESPONSE to REVIEWER COM-MENT –Section 5 We have attempted to model resilience using three types of models: 1) a descriptive model that outlines our abstract interpretation of community resilience as a system; 2) a mathematical model equivalent of 1 illustrated using geometric reasoning; and 3) a fuzzy logic equivalent of 2 for the purpose of computational analysis in the face of limited and subjective data.

REVIEWER'S COMMENT –Section 6 What is the source of the resilience input factors? Were the inputs verified to see if the model worked? In the "hypothetical" analysis who determined the inputs (e.g. who did the assessment as to the values of the inputs)? There is no explanation of this in Section 4 Model Application, just a very generic text about the study location. When the "results" appear, they are more like a description of the tool and how it can be used rather than results based on empirical and/or qualitative assessments. Thus, the information presented in the manuscript does not

support the results as presented. In addition, the discussion and conclusion section is not especially robust either and in many ways rehashes the literature review rather than presenting new and innovative findings related to resilience in flood prone communities. This paper could be significantly improved by re-framing it as a methodological contribution where the conceptual model and its mathematical expression is more fully articulated including all the requisite input variables including the sources. Then the fuzzy logic scoring template/tool can be described in more detail. In order to test the model, however, the authors would need to generate at least a small sample of stakeholders to complete the input variable assessments as a measure of the validity of the effort. This is a difficult paper to assess given how much of it seems focused on the modeling (Figures 2-6) and recovery quality, yet in these same figures there's no mention of the other two components (resource availability and resource utilization processes) unless these are both subsumed under resources per Figure 3. As a reader I do not understand the model and its conversion to a type of resilience index (the stated output). Whether this is a function of my lack of familiarity with mathematical modeling as used here or the authors' explanation of it is uncertain. Either way, the manuscript needs a rewrite to make it appeal more directly to the journal's readership. AUTHORS' RESPONSE to REVIEWER COMMENT -Section 6 An extensive literature search was the basis for identifying the input variables/ factors. The whole essence of adopting a soft computing tool, fuzzy logic, is to enable subjective opinions and limited data to be summarized using linguistic variables as input into the inference system. A fuzzy inference system/ model of resilience is a template that allows experts and other stakeholders to translate their perceptions of the problem and map their linguistics rating of these variables into an index based on the fuzzy computational relationships we have defined. This will be emphasized in the revision. Our sample application was based on the outcome of field study, reflective interactions with experts, and stakeholders familiar with study locations. Our sample scoring was therefore based on our interactions with these various stakeholders, which include academics, community leaders, and our understanding of their opinions, as well as the data extracted from various

historical records. We will make efforts to improve on the explanations for the readers.

Reference Cutter, S. L., Barnes, L., Berry, M., & Burton, C. (2008). A place-based model for understanding community resilience to natural disasters. Global environmental change , 18 (4), 598-606. Keating, A., Campbell, K., Szoenyi, M., McQuistan, C., Nash, D., & Burer, M. (2017). Development and testing of a community flood resilience measurement tool. Natural Hazards and Earth System Sciences. 2017;17(1):77 , 17 (1), 77.

DROP model reproduced from Cutter et al 2008

Fig. 1. DROP Model

---

## Author Response (AR1)

REVIEWER COMMENTS
AUTHORS' RESPONSE/REVISIONS/

**REVIEWER's COMMENT –Section 1**a
This paper adds to expanding literature on disaster resilience measurement. The primary purpose of the study is to develop a mathematical model based on the U.S. National Academies definition of resilience ("the ability to prepare and plan for, absorb, recover from, and more successfully adapt to adverse events") and then implement the model for three flood-prone communities using a fuzzy logic equivalent.

**AUTHORS' RESPONSE to REVIEWER COMMENT –Section 1a**
Comments reflect the broad scope of the paper

**REVIEWER's COMMENT –Section 1b**
The background on the development of operational resilience measurement models is good, although it does rely primarily on relatively few papers (e.g. Cai et al. 2018; Cutter 2018; Keating et al. 2017; Zou et al. 2018), and perhaps misses other community resilience measurement efforts such as Zurich's Flood Resilience program.

**AUTHORS' RESPONSE to REVIEWER COMMENT –Section 1b**
While efforts will be made to include other relevant papers, the authors wish to state that the Zurich flood resilience approach was considered through one of the papers, Keating et al, 2017 summarizing the Zurich resilience measurement. We have revised the text to acknowledge explicitly the efforts of the Zurich program.

**REVIEWER's COMMENT –Section 2a**
I am also concerned that the definitional discourse does not adequately describe the complexities and variability in the meaning of resilience as it is applied to a particular system, event, or more broadly to capture community abilities as the NRC definition is designed to do?

**AUTHORS' RESPONSE to REVIEWER COMMENT –Section 2a**
We revised Section 1 significantly and beefed up the discussion on definition convergence and related complexities and to reflect the dynamic components of the community resilience system

**REVIEWER's COMMENT –Section 2b**
I would encourage the authors to reduce the definitional discussion and simply select and then justify the definition they prefer to use (e.g. NRC 2012) as the basis for their conceptual model.

**AUTHORS' RESPONSE to REVIEWER COMMENT –Section 2a**
We revised the introduction to enhance the flow of thoughts around the definitional issues, but believe they are critical to understanding the work that follows in the paper.

**REVIEWER's COMMENT –Section 2b**
In the formulation of the conceptual mode (Figure I,) the authors assume that resilience leads to recovery (the outcome of interest). How does the conceptual model line up with their preferred definition? While an attempt was made on p. 5-6 to do this, most of the discussion is focused on recovery or the recovery spectrum. So, how can the operationalization of a definition that includes recovery also be used to measure an outcome, also labeled recovery?

**AUTHORS' RESPONSE to REVIEWER COMMENT –Section 2b**
We have improved the clarity of Figure 1, firstly by using a two way arrow arc to depict the interaction between resilience and recovery and secondly by enhancing the explanation of the figure with respect to the proposed model.   We say our model is our interpretation of the definition as well as our understanding of the interactions related to resilience. Our schematic model for instance recognizes that resilience enhances recovery and/or that recovery is an outcome of resilience whereby when a community, as coupled system, becomes more resilient its capacity to experience post disaster increases. In other words recovery, in terms of time taken to attain post disaster recovery and the degree of recovery attained are influenced by the resilience. This understanding is supported by the DROP resilience model as illustrated in (Cutter, Barnes, Berry, & Burton, 2008) which we have added to the manuscript.

[Figure]

Schematic representation of the disaster resilience of place (DROP) model.

DROP model reproduced from  Cutter et al 2008

Our model implicitly suggests that recovery (i.e. recovery time and quality) can surrogate resilience. This is reasonable because post disaster recovery is driven by inherent resilience factors some of which we further explained in Table 1 of this paper

**REVIEWER's  COMMENT –Section 3**
The authors need to clearly distinguish resilience (an outcome in and of itself) from recovery or at a minimum more clearly articulate they are describing resilience-type capacities within communities that influence flood recovery. It seems to me that the conceptual model is oriented to flood recovery (p. 7) rather than resilience per se. Later on in the paper, they use the resilience index as the output (Table 6), but this is not found in the conceptual model as described in Figure 1.

**AUTHORS' RESPONSE to REVIEWER COMMENT –Section 3**
It should be noted that this reviewer's comment underpins our argument about how the absence of consensus on definition leads to divergent interpretations of the interactions among the components of the resilience system. According to (Cutter, Barnes, Berry, & Burton, 2008) multiple definitions of resilience exist within the literature, with no broadly accepted single definition. Our schematic model for instance recognizes that resilience enhances recovery and/or that recovery is an outcome of resilience whereby when a community, as coupled system, becomes more resilient its capacity to experience post disaster increases. As noted above, we have added the DROP model and additional discussion to the paper to support our argument.

**REVIEWER's  COMMENT –Section 4**
What is unique about the context of flood hazards in the model, or could it equally apply to any natural hazard impact in a community?

**AUTHORS' RESPONSE to REVIEWER COMMENT –Section 4**
The focus of the model is flood hazards, but flood hazards share characteristics with other natural hazards. Most of the factors provided in Table 1 apply to any hazard, so the model would be applicable.

**REVIEWER's  COMMENT –Section 5**
The bulk of the paper describes the mathematics of the model and its implementation, but again I wonder as to whether the model is describing and/or modeling resilience.

**AUTHORS' RESPONSE to REVIEWER COMMENT –Section 5**
We have attempted to model resilience using three types of models: 1) descriptive model that outlines our abstract interpretation of the community resilience as a system, 2) a mathematical model equivalent of 1 illustrated using geometric reasoning, and 3) a fuzzy logic equivalent of 2 for the purpose computational analysis in the face of limited and subjective data. We believe application of the model provides a template for measuring resilience, which is one of our objectives.

**REVIEWER's COMMENT –Section 6**

What is the source of the resilience input factors? Were the inputs verified to see if the model worked? In the "hypothetical" analysis who determined the inputs (e.g. who did the assessment as to the values of the inputs)?

There is no explanation of this in Section 4 Model Application, just a very generic text about the study location. When the "results" appear, they are more like a description of the tool and how it can be used rather than results based on empirical and/or qualitative assessments. Thus, the information presented in the manuscript does not support the results as presented. In addition, the discussion and conclusion section is not especially robust either and in many ways rehashes the literature review rather than presenting new and innovative findings related to resilience in flood prone communities. This paper could be significantly improved by re-framing it as a methodological contribution where the conceptual model and its mathematical expression is more fully articulated including all the requisite input variables including the sources.

Then the fuzzy logic scoring template/tool can be described in more detail. In order to test the model, however, the authors would need to generate at least a small sample of stakeholders to complete the input variable assessments as a measure of the validity of the effort. This is a difficult paper to assess given how much of it seems focused on the modeling (Figures 2-6) and recovery quality, yet in these same figures there's no mention of the other two components (resource availability and resource utilization processes) unless these are both subsumed under resources per Figure 3. As a reader I do not understand the model and its conversion to a type of resilience index (the stated output). Whether this is a function of my lack of familiarity with mathematical modeling as used here or the authors' explanation of it is uncertain. Either way, the manuscript needs a rewrite to make it appeal more directly to the journal's readership.

**AUTHORS' RESPONSE to REVIEWER COMMENT –Section 6**

An extensive literature search was the basis for identifying the input variables/ factors. The whole essence of adopting a soft computing tool, fuzzy logic, is to enable subjective opinions and limited data to be summarized using linguistic variables as input into the inference system. A fuzzy inference system/ model of the resilience is a template that allow experts and other stakeholders to translate their perceptions of the problem and map their linguistics rating of these variables into index based on the fuzzy computational relationships we have defined. Our sample application was based on the outcome field study, reflective interactions with experts and stakeholders familiar with study locations. The sample scoring was therefore based on the opinions of these various stakeholders, as well as data extracted from various historical records. This has been added to the text in Section 4.1.

This manuscript contributes to broad research field on community resilience and aims to develop framework on measuring resilience. After examining and discussing the challenges of different definition and concepts in this context, the authors presented a conceptual and mathematical model as well as applied a fuzzy logic approach to generate a resilience index, which was applied in three flood-prone communities in the US (North Carolina and Virginia).

**AUTHORS' RESPONSE to REVIEWER COMMENT –Section 1**
Comments reflect the broad scope of the paper

**REVIEWER's COMMENT –Section 2a**
The manuscript is in general well written but the structure and the different level of information provided in the sections challenge the reader to follow the argumentation of authors and relate the different parts of the framework.

**AUTHORS' RESPONSE to REVIEWER COMMENT –Section 2a**
We have noted this useful observation. We improved on the structure and level of information to enhance overall flow of our argument and readability by the target audience

**REVIEWER's COMMENT –Section 2b**
For example, the introduction provides a selected overview on the topic and challenges of community resilience and different frameworks to measure resilience. However, the focus is on different definition of resilience and not on the differences in approaches to measure resilience, which are only mentioned but not explain. I see here a high potential to reduce the definition discussion and provide more details on measuring resilience.

**AUTHORS' RESPONSE to REVIEWER COMMENT –Section 2b**
We appreciate the need to beef up discussions on existing measuring approaches as well as their differences. We included further discussion and literature on the differences in measuring approaches. However, we decided to retain our current discussion on definitions with some modifications that relate to differences in measuring approaches. We also added to the discussion on approaches.

**REVIEWER's COMMENT –Section 3**

Moreover, I suggest to present also a clear objective for the study, which would help to follow the structure of the manuscript. Perhaps a flow chart showing the interrelation of the different models would also increase the understanding of the chosen structure.

**AUTHORS' RESPONSE to REVIEWER COMMENT –Section 3**

We now provide a specific section on aims and objectives.

**REVIEWER's COMMENT –Section 4a**

Furthermore, in the design of the model (also including Fig. 1) it is not very well explained why resilience leads than to recovery, as one part of resilience would be 'how the community is able to recover?' and thus this parameter should contribute to measure resilience.

**AUTHORS' RESPONSE to REVIEWER COMMENT –Section 4a**

We appreciate the need to improve the clarity of Figure 1, firstly by using a two-way arrow arc to depict the interaction between resilience and recovery and secondly by enhancing the explanation of the figure with respect to the proposed model.

It should be noted that this reviewer's comment underpins our argument about how the absence of consensus on definition leads to divergent interpretations of the interactions among the components of the resilience system. According to Cutter, Barnes, Berry, & Burton (2008), multiple definitions of resilience exist within the literature, with no broadly accepted single definition. Our schematic model recognizes that resilience enhances recovery and/or that recovery is an outcome of resilience whereby when a community, as a coupled system, becomes more resilient its capacity to experience post disaster recovery increases. In other words, recovery, in terms of the time taken to attain post disaster recovery and the degree of recovery attained are influenced by the resilience. This understanding is supported by the DROP resilience model as illustrated in Cutter, Barnes, Berry, & Burton (2008).

Thus, our model implicitly suggests that recovery (recovery time or quality) can be a substitute for resilience. This is reasonable because post disaster recovery is driven by factors that characterize resilience

**REVIEWER's COMMENT –Section 4b**

The authors illustrated in section 2.5 some extreme cases and showed the gained insights to model structure, however it also shows the limitation of the model regarding

dynamical change, e.g. if you consider in case 1 that you have no efficiency in the resource utilization processes = no resilience, then your model ignores any preparedness

and coping capacity. I would agree on a long term but you measuring only in static

manner, thus should not there be a difference of communities with different hazard

absorbing capacity?

**AUTHORS' RESPONSE to REVIEWER COMMENT –Section 4b**

The model does not ignore preparedness and coping capacity. Rather the 'extreme' scenarios were used so as to demonstrate the nature of the model's 3 consolidated dimensions of Hazard absorbing capacity, Resource use processes, and Resource availability. Note that from Table 1, these 3 main dimensions are each functions of several resilience factors. For instance,

preparedness is one the factors or components of the resource use system (or process efficiency or community governance processes) simply termed efficiency, while coping capacity is one of the factors captured in the dimension of Hazard absorbing capacity. We have reworked the discussion which we hope makes it more clear.

**REVIEWER's COMMENT –Section 5**
The authors are also encourage to provide more thoughts about
their assumption of that 'negative' resilience is another expression of vulnerability.
**AUTHORS' RESPONSE to REVIEWER COMMENT –Section 5**
We note this observation and therefore adopt a clearer expression to avoid misinterpretation. The idea being conveyed is that the bulk of the 'resilience area' lies in the low (negative) quadrant when the hazard absorbing (Coping) and governance process/resource use efficiency deteriorate. Note that the absorbing capacity encompasses social, infrastructural, technical, and psychological factors that determine system's vulnerability. The concept of negative resilience has been revised as a 'resilience reservoir quadrant' and further explanation is provided to link with vulnerability.

**REVIEWER's COMMENT –Section 6**
The structure of section 3 is confusing because in the beginning it is not clear why the fuzzy logic is addressed and how it is related to the previous sections. Furthermore, from 3.2. onwards more detailed information on chosen criteria for selecting variables, number of rules, type of membership functions, weights : : : are needed. Currently, in this section a lot of questions arise, but I see a high potential to improve the whole manuscript if you revise this section (see detailed remarks in the attached file).
**AUTHORS' RESPONSE to REVIEWER COMMENT –Section 6**
After developing a mathematical model, the next logical step is model analysis or solution method. We have adopted the fuzzy inference system as the mathematical/computational tool for analyzing the resulting model. The objective section is to develop the fuzzy inference equivalent of the model. We included more detailed information on the fuzzy logic rules and weights.

**REVIEWER's COMMENT –Section 7**
In general the application in case study is only a very vague description and it is not clear on which assumption you based your hypothetical input score.
**AUTHORS' RESPONSE to REVIEWER COMMENT –Section 7**
The data we used were actual real life data. Maybe the phrase' hypothetical input score' may not have been the best to use. The process of data gathering and sample scoring is now explicitly explained in Section 4.1 to show that data used was based on real life situations. Our sample application was based on the outcome of field study, reflective interactions with experts, and stakeholders familiar with the study locations. Our sample scoring was therefore based on our interactions with these various stakeholders, which include academics, community leaders, and our understanding of their opinions, as well as the data extracted from various historical records. For instance. at Windsor during a planners' conference that brought together academics, officers from state and federal agencies dealing with emergency

management, community leaders, and officers of the towns, we gained useful insights on flood resilience activities. Similarly, the authors visited Norfolk VA and took a tour of the city under the guide of GIS experts from one of the local universities.  These interactions and associated field study were used to generate the sample scoring.

**REVIEWER's  COMMENT –Section 8**

 I also would see an added value - if you stay with hypothetical inputs - to gain more insights on the sensitive of your model with systematically testing of different input data but also on the rule setting and membership functions. The added-value would be a better understanding of model. The discussion and conclusion is very generic and needs to be rewritten regarding the points highlighted in the introduction, the (missing) objectives and the gained insights. I indicated different ways how the authors may restructure and rewrite the manuscript to show the added value of this study to the readers and scientific community. See detailed comments in the attached file.

**AUTHORS' RESPONSE to REVIEWER COMMENT –Section 8**

These observations are noted, the discussion was revised to align with both the reviewers' comments and the objectives stated earlier.

1. We edited the document throughout for grammar and typographical errors.
2. The abstract was revised to be more explicit about the objectives, methods, and results
3. Section 1.0
   a. We deleted some paragraphs and moved others to enhance the flow
   b. We rewrote several sentences to enhance clarity
   c. We added a discussion of the Zurich Alliance approach
   d. We added to the discussion of the multiplicity of definitions of resilience to strengthen the foundation for the work
   e. We rewrote the ending of the section to provide background for the methodology we used
4. Section 1.1
   a. We added this section to be more explicit about aims and objectives
5. Section 1.2
   a. We moved this discussion on fuzzy logic to the introduction to et the stage for the work that follows
6. Section 2.1
   a. We rewrote the discussion of the conceptual framework
   b. We revised Figure 1 based on reviewer comments
   c. We added Figure 2 to provide support for our model
7. Section 2.2.1
   a. The presentation of terms, notations, and definitions is now consolidated to enhance flow and understanding.
   b. What was Table 3 is now Table 1 – renumbered for better structure
8. Section 2.2.3
   a. We revised the discussion of negative resilience and provided additional explanation
9. Section 4.0
   a. We rewrite this section to make our data collection methods more clear
10. Section 5.0
    a. The discussion and conclusions have been significantly revised to reflect the objectives stated in the Introduction.

[revised manuscript text omitted]

framework (Szoenyi, et al., 2016) (Keating et al. 2017). This model which has evolved through intensive use of case studies of diverse flooded communities, however, requires trained resilience assessors to grade sources of resiliences based on a technical risk grading standard (TRGS) developed by Zurich risk experts.

Commented [MBM5]: Added to provide more information and enhance discussion

Despite the attention resilience has gained, the concept remains difficult to operationalize in the context of community flood risk management due to, among other factors, the difficulty in measuring resilience (Cutter, 2018; Fisher, 2015). Many experts and authors have noted the difficulty in integrating indicators of the natural and human systems as well as socio-environmental factors into resilience by most of the existing frameworks (Cai et al., 2018; Cutter, 2018; Fuchs and Thaler, 2018; Qiang and Lam, 2016). Resilience, as a multifaceted and multidimensional concept, has developed across multiple disciplines and applications such that resilience discourse has attracted multidisciplinary interests from both research and policy perspectives. While the wide spectrum of multidisciplinary and practice interests characterizing resilience discourse has increased its understanding and generated insights, it has also led to the emergence of multiple variants of its definiton as well as the absence of consensus on the conceptual framework for its measurement (Brown and Williams, 2015; Cohen et al., 2016; Cutter 2018). For instance, resilience has been noted to have varied definitions depending on the hazard and disciplinary contexts, with over 70 definitions identified by Fisher (2015).

Commented [MBM6]: This paragraph recast and beefed up to moderate the discussion on definition convergence

[revised manuscript text omitted]

There are two commonly used fuzzy inference systems: the Mamdani-type and Sugeno-type. While the Sugeno systems offer more compact and computationally efficient representations, the Mamdani systems are however more intuitive, have widespread acceptance and are well-suited to human input (Oladokun and Emmanuel, 2014). The Mamdani FIS will behas been adopted for this study. The FIS is characterized by the use of linguistic variables and their term sets, the membership functions for the fuzzification and de-defuzzification processes, and the fuzzy rules.

The concept of membership function (MF) is central to FIS. In traditional logic, an element $x$ is either in or out of crisp set A; in other words, its degree of membership of the set is either zero or one. However, in fuzzy logic the element $x$ can be in a fuzzy set B 'partially' by using a MF $\mu_B(x)$ which can return any real value between 0 and 1. This returned value is the degree of membership representing the degree to which the element belongs to a fuzzy set. Therefore, in FL, the truth of any statement becomes a matter of degree.

Thus for crisp set A $\quad \mu_A(x) = \begin{cases} 1 & if\ x\ \in A \\ 0\ otherwise \end{cases}$

On the other hand, for a fuzzy set, the MF may be represented as follows

[revised manuscript text omitted]
 [Book Section] // Vulnerability and resilience to natural hazards / book auth. Fuchs S and Thaler T. - [s.l.] : Cambridge University Press, 2018.

**Filion P and Sands G** Enhancing Hazard Resilience among Impoverished Urban Communities in Ghana: The Role of Women as Catalysts for Improvement [Book Section] // In Cities at Risk: Planning for and Recovering from Natural Disasters. - [s.l.] : Routledge., 2016.

**Fisher L** Disaster responses: More than 70 ways to show resilience [Journal] // Nature. - 2015. - 7537 : Vol. 518. - pp. 35-35.

**Folke C [et al.]** Resilience and sustainable development: building adaptive capacity in a world of transformations [Journal] // AMBIO: A journal of the human environment. - 2002. - 5 : Vol. 31. - pp. 437-440.

**Folke C** Resilience:theemergenceofaperspectivefor social–ecologicalsystemsanalyses [Journal] // GlobalEnvironmental Change. - 2006. - Vol. 16. - pp. 253–267..

**Fuchs S and Thaler T** Vulnearabilty and Resilience to Natural Hazards [Book]. - New York : Cambridge Press, 2018.

**Guo L [et al.]** A comprehensive flash flood defense system in China: overview, achievements, and outlook [Journal] // Nat Hazards. - 2018. - pp. 1-14.

**Hammond M J [et al.]** Urban flood impact assessment: A state-of-the-art review [Journal] // Urban Water Journal. - 2015. - 1 : Vol. 12. - pp. 14-29.

**Ibanez G E [et al.]** Qualitative Analysis of Coping Strategies Among Mexican Disaster Survivors [Journal] // Anxiety, Stress, and Coping. - [s.l.] : Taylor and Francis , 2004. - 1 : Vol. 17. - pp. 69-85.

**Jha A K, Bloch R and Lamond J** Cities and Flooding: A guide to Integrated Urban Flood Risk Management the 21st Centuries [Report]. - Washington : The World bank, 2012. - p. 631.

**Joseph R, Proverbs D and Lamond J** Resilient reinstatement: what can we learn from the 2007 flooding in England? [Book Section] // Flood Recovery, Innovation and Response IV / book auth. Proverbs D and Brebbia C A. - [s.l.] : WIT Press, 2014. - Vol. 184.

**Keating A [et al.]** Development and testing of a community flood resilience measurement tool [Journal] // Natural Hazards and Earth System Sciences. 2017;17(1):77. - 2017. - 1 : Vol. 17. - p. 77.

**Kron W** Flood Risk = Hazard • Values • Vulnerability [Journal] // Water International. - 2005. - 1 : Vol. 30. - pp. 58-68.

**Lee A V, Vargo J and Seville E** Developing a Tool to Measure and Compare Organizations' Resilience [Journal] // Natural Hazards Review. - 2013. - 1 : Vol. 14. - pp. 29-41.

**Lincy G R M and John C J** A multiple fuzzy inference systems framework for daily stock trading with application to NASDAQ stock exchange [Journal] // Expert Systems with Applications: An International Journal. - 2016. - C : Vol. 44. - pp. 13-21..

**Mallakpour I and Villarini G** The changing nature of flooding across the central United States [Journal] // Nature Climate Change. - 2015. - Vol. 5. - pp. 250–254.

**Mamdani E H and Assilian S** An experiment in linguistic synthesis with a fuzzy logic controller [Journal] // International Journal of Man-Machine Studies. - 1975.. - 1 : Vol. 7. - pp. 1-13.

**Mavhura E [et al.]** Indigenous knowledge,coping strategies and resilience to floods in Muzarabani,Zimbabwe [Journal] // International Journal of Disaster Risk Reduction. - 2013. - Vol. 5. - pp. 38-48.

**Montz B** Emerging Issues and Challenges: Natural Hazards [Journal] // Journal of Contemporary Water Research & Education. - 2009. - Vol. 142. - pp. 42-45.

Multi-dimensional hurricane resilience assessment of electric power systems [Journal] // Structural Safety. - 2014. - Vol. 48. - pp. 15-24.

**Norris F H [et al.]** Community resilience as a metaphor, theory, set of capacities and strategy for disaster readiness [Journal] // Community Psychology. - 2008. - Vol. 41. - pp. 127-50.

**NRC** Disaster resiience - a national imperitive [Report] = . - Washington, DC : National Academies press, 2012.

**Oladokun V O and Emmanuel C G** Urban Market Fire Disasters Management in Nigeria: A Damage Minimization based Fuzzy Logic Model Approach [Journal] // International Journal of Computer Applications. - 2014. - 17 : Vol. 106.

**Oladokun V O, Proverbs D G and Lammond J** Measuring flood resilience: A fuzzy logic approach [Journal] // International Journal of Building Pathology and Adaptation. - 2017.

**Park J [et al.]** Integrating risk and resilience approaches to catastrophe management in engineering systems [Journal] // Risk Analysis. - 2013. - 3 : Vol. 33. - pp. 356-367.

**Parsons M [et al.]** Top-downassessmentofdisasterresilience:Aconceptualframework using copingandadaptivecapacities [Journal] // InternationalJournalofDisasterRiskReduction. - 2016. - Vol. 19. - pp. 1-11.

**Qiang Y and Lam N S N** The impact of Hurricane Katrina on urban growth in Louisiana: An analysis using data mining and simulation approaches. [Journal] // International Journal of Geographical Information Science. - 2016. - 9 : Vol. 30. - pp. 1832–52.

**Rose A** Broader Dimensions of Economic Resilience [Book Section] // Defining and Measuring Economic Resilience from a Societal, Environmental and Security Perspective. Integrated Disaster Risk Management. - SSingapore : Springer, 2017.

**Schelfaut K [et al.]** Bringing floodresilienceintopractice:theFREEMAN project [Journal] // Environmental Science & Policy. - 2011. - 7 : Vol. 14. - pp. 825-833.

**Shah A A, Ye J and Abid M** Flood hazards: household vulnerability and resilience in disaster-prone districts of Khyber Pakhtunkhwa province, Pakistan [Journal].

**Sharifi A** A critical review of selected tools for assessing community resilience [Journal] // Ecological Indicators. - 2016. - Vol. 69. - pp. 629–647.

**Su Y S** Discourse, Strategy, and Practice of Urban Resilience against Flooding [Journal] // Business and Management Studies. - 2016. - 1 : Vol. 2. - pp. 73-87.

**Su Y S** Urban Flood Resilience in New York City, London, Randstad, Tokyo, Shanghai, and Taipei [Journal] // Journal of Management and Sustainability. - 2016. - 1 : Vol. 6. - p. 92.

**Swalheim. S and Dodman D** Building resilience: how the urban poor can drive climate adaptation [Report]. - London : IIED, 2008.

**Thomas J A and Mora K** Community resilience, latent resources and resource scarcity after an earthquake: Is society really three meals away from anarchy? [Journal] // Natural hazards. - 2014. - 4 : Vol. 72. - pp. 477-490..

**Tompkins E and Adger W N** Does adaptive management of natural resources enhance resilience to climate change? [Journal] // Ecology and society. - 2004. - 2 : Vol. 9.

**Trogrlić R Š [et al.]** Taking stock of community-based flood risk management in Malawi: different stakeholders, different perspectives [Journal] // Environmental Hazards . - 2018. - 2 : Vol. 17.

**Wing O E J [et al.]** Estimates of present and future flood risk in the conterminous United States [Journal] // Environmental Research Letters. - 2018. - Vol. 13.

**Zadeh L A** Fuzzy logic= computing with words [Journal] // Fuzzy Systems, IEEE Transactions on. - 1996. - 2 : Vol. 4. - pp. 103-111..

**Zou L [et al.]** Mining Twitter Data for Improved Understanding of Disaster Resilience [Journal] // Annals of the American Association of Geographers. - 2018. - pp. 1–20.

[Figure]

Figure 1: Resilience measuring conceptual framework

[Figure]

Schematic representation of the disaster resilience of place (DROP) model.

Figure 2: The DROP model reproduced from Cutter et al., 2008

[Figure]

Figure 3: Resilience conceptual model

[Figure]

Figure 4: Resilience area = 0 when  ρ= Sin Θ= 0

[Figure]

Fig. 5: Resilience area ($a_i = hg$) maximizes recovery resources g on absorbing capacity h

[Figure]

Figure 6: Resilience as Absorbing Capacity approaches zero

[Figure]

Figure 7 Resilience fuzzy inference systems

[Figure]

Figure 8. Resilience output surface plots.

[Figure]

Figure 9. The study area

[Figure]

Figure 10: Rule setting and output Greenville

---

## Referee Report (RR1)

The manuscript develops a conceptual model for assessing flood resilience and applies it to three US cities. The objective is clear, and the manuscript is generally well-organized and easy to follow. It has good potential to contribute to the growing body of global literature on flood resilience. The following comments could be viewed as a way to improve the presentation quality of the manuscript.

1. The introduction section could be shorter. The authors spent much space to describe the background information and methods at the expense of explaining the potentially interesting results of the authors' case studies and discussing the results.

2. While generally well-written, the authors repeated the same idea at places, which I think could be trimmed for brevity. Considering the manuscript is a little bit longer than a typical manuscript, it needs to be condensed without losing the main points of the study. I suggest that the authors consider moving some description of the model section 2 into an appendix (e.g., section 2.2.2). See my remarks below.

3. In general, the methods are well-described with sufficient details in most places, but the authors could state how the exact weight was derived in more detail. The authors stated that "*the sample scoring was based on the insights derived from our understanding of their opinions, as well as data extracted from various historical records*." How did the authors quantify the diverse opinions of various stakeholders? How many stakeholders were consulted? What were the selection criteria of these stakeholders the selection of historical records since there could be many different stakeholders and many different historical records (policy documents, newspaper articles, etc.)? Did the author use any specific technique to derive weighted averages (e.g., AHP)? As it stands, it is a little bit difficult to understand how the authors did and replicate how the authors' methods.

Other comments

Abstract: It is not typical to cite a reference in the abstract.

Page 2, lines 36-37. Lines 51-52. Similar ideas are repeated twice.

Pages 2-3, lines 71-90. This information could be omitted or create a table replacing the text.

Page 3, line 61. Insert comma after "Furthermore".

Page 5, line 118. Insert comma after "Therefore".

Page 6, line 162. Remove "that" before "that"

Page 6, lines 167-172. This paragraph could be omitted.

Pages 7-8. Section 2.1 This section could be condensed as similar ideas are stated repeated at multiple times. (e.g., definition of resilience, a three factor reservoir system – repeated later in section 2.2)

Page 9, line 233. Insert comma after "In other words".

Page 13, lines 322 to 328. Already explained in Table 1.

Page 15, lines 388-389. Already explained before.

Page 16, line 398. Be consistent using small or large capitals (e.g., moderate vs. Moderate).

Page 16, lines 412-415. This sentence could be omitted.

Page 18, line 441. "flat topography". Why not reporting slope of each city in Table 5?

Page 18, line 447. "they have rather different flood regimes and histories". It would be interesting to see how flood regimes and histories are different across the cities.

Page 18, Table 5. Some units are missing (e.g., median income, US citizenship, mean property value)

Page 19, line 457-458. "data extracted from various historical records" what data are you referring to?

Page 19, line 458. Insert comma after "For instance"

Page 19, line 463-464. Could the authors elaborate this statement a little bit further?

Page 19, line 471-472. Are those numbers normalized by population size?

Page 20, lines 490-491. It would be interesting if the authors interpret the results of the output more in the context of flood resilience research.

Page 21, line 521-524. Not sure if the authors confirm the need from their study findings.

---

## Editor Decision (ED1)

[revised manuscript text omitted]

Hazard absorbing $H$

$\Theta= 0$

$a_i=0$

g

Recovery Quality $Q$

Figure 4: Resilience area = 0 when $\rho=$ Sin $\Theta= 0$

[Figure]

Fig. 5: Resilience area ($a_i$ = hg) maximizes recovery resources g on absorbing capacity h

[Figure]

Figure 6: Resilience as Absorbing Capacity approaches zero

[Figure]

Figure 7 Resilience fuzzy inference systems

[Figure]

Figure 8. Resilience output surface plots.

[Figure]

Figure 9. The study area

[Figure]

Figure 10: Rule setting and output Greenville

---

## Author Response (AR2)

The manuscript develops a conceptual model for assessing flood resilience and applies it to three US cities. The objective is clear, and the manuscript is generally well-organized and easy to follow. It has good potential to contribute to the growing body of global literature on flood resilience. The following comments could be viewed as a way to improve the presentation quality of the manuscript.

1. The introduction section could be shorter. The authors spent much space to describe the background information and methods at the expense of explaining the potentially interesting results of the authors' case studies and discussing the results.

*Authors' Response – We have reduced the introduction by recasting or removing some section: information still adequate*

2. While generally well-written, the authors repeated the same idea at places, which I think could be trimmed for brevity. Considering the manuscript is a little bit longer than a typical manuscript, it needs to be condensed without losing the main points of the study. I suggest that the authors consider moving some description of the model section 2 into an appendix (e.g., section 2.2.2). See my remarks below.

*Authors' Response -1. Citation removed from abstract*

*2. Section 2.2.2 has been considerably reduced although section 2.2.2 has not been moved to appendix*

3. In general, the methods are well-described with sufficient details in most places, but the authors could state how the exact weight was derived in more detail. The authors stated that "*the sample scoring was based on the insights derived from our understanding of their opinions, as well as data extracted from various historical records.*" How did the authors quantify the diverse opinions of various stakeholders? How many stakeholders were consulted? What were the selection criteria of these stakeholders the selection of historical records since there could be many different stakeholders and many different historical records (policy documents, newspaper articles, etc.)? Did the author use any specific technique to derive weighted averages (e.g., AHP)? As it stands, it is a little bit difficult to understand how the authors did and replicate how the authors' methods.

*Authors' Response: The fuzzy model use of linguistic variables to capture input quantities allows experts' opinion to be captured fairly well. The stakeholders were selected from those who are familiar with issues of flooding.*

Other comments
Abstract: It is not typical to cite a reference in the abstract.
*Authors' Response -Citation removed from abstract*

Page 2, lines 36-37. Lines 51-52. Similar ideas are repeated twice.
*Authors' Response – This repetition has been corrected as line 36-37 recast to accommodate the 2 similar ideas*

Pages 2-3, lines 71-90. This information could be omitted or create a table replacing the text.

*Authors' Response – We believe this information adds some context to the discussion hence we suggest its retention in the current format*

Page 3, line 61. Insert comma after "Furthermore".
    *Authors' Response – comma inserted after "Furthermore*
Page 5, line 118. Insert comma after "Therefore".
    *Authors' Response – comma inserted after "Therefore"*

Page 6, line 162. Remove "that" before "that"
    *Authors' Response- "that" removed*
Page 6, lines 167-172. This paragraph could be omitted.
    *Authors' Response-  Paragraph omitted – while the information on  type fuzzy inference system used  mentioned in passing in line 195.*

Pages 7-8. Section 2.1 This section could be condensed as similar ideas are stated repeated at multiple times. (e.g., definition of resilience, a three factor reservoir system – repeated later in section 2.2)
    *Authors' Response- we believe this section should remain close to as it is for clarity, but some of the repetition was removed.*

Page 9, line 233. Insert comma after "In other words".
    *Authors' Response – comma inserted after "In other words".*

Page 13, lines 322 to 328. Already explained in Table 1.
    *Authors' Response –  Repetition  removed*

Page 15, lines 388-389. Already explained before.
    *Authors' Response –  lines deleted*

Page 16, line 398. Be consistent using small or large capitals (e.g., moderate vs. Moderate).
    *Authors' Response- capital adopted*

Page 16, lines 412-415. This sentence could be omitted.
        *Authors' Response –  lines deleted*

Page 18, line 441. "flat topography". Why not reporting slope of each city in Table 5?
    *Authors' Response : we have included elevation of each city in table 5 to characterize topography; we do not have slope for the 3 cities*

Page 18, line 447. "they have rather different flood regimes and histories". It would be interesting to see how flood regimes and histories are different across the cities.
    *Author's Response: A short explanation of regimes was added*

Page 18, Table 5. Some units are missing (e.g., median income, US citizenship, mean property value)
        *Authors' Response – missing units included*

Page 19, line 457-458. "data extracted from various historical records" what data are you referring to?

*Authors' Response-   sentence recast to read   " .. demographic and  socio-economic information extracted from various historical records*.'

Page 19, line 458. Insert comma after "For instance"
    *Authors' response -- comma inserted after "For instance"*

Page 19, line 463-464. Could the authors elaborate this statement a little bit further?
        *Authors' Response:  further elaboration on  statement  provided*
Page 19, line 471-472. Are those numbers normalized by population size?
        *Authors' Response – numbers not normalized by population size*

Page 20, lines 490-491. It would be interesting if the authors interpret the results of the output more in the context of flood resilience research.
*Authors' Response : we attempt to provide further information in the* context of flood resilience research

Page 21, line 521-524. Not sure if the authors confirm the need from their study findings.

*Authors' Response – this statement is suggesting one of the potential applications of the model .. such application was not part of the study : the statement has been deleted to avoid ambiguity ..*

Towards Measuring Resilience of Flood Prone Communities: A Conceptual Framework

[revised manuscript text omitted]

---

## Author Response (AR3)

NHESS-2018-217

Towards Measuring Resilience of Flood Prone Communities: A Conceptual Framework

Authors response

Page 3 line 69: citation format corrected

Line 74:  DFID written in full

Page 5 line 119: citation format corrected

Page 6 lines 164-165: References provided

Page 8 line 211: citation format corrected

Page 10 line 263:  Table 1 formatted according to journal requirements , and moved to a separate sheet

Page 12 line 301: '3'  written in words

Page 13 line 344: GUI is defined

Page 13   lines 344-347:  The response to (reviewer 2, comment to section 7)  has been provided in the section 4.1 of the revised  manuscript ( see page  18 line 433-435)

Page 14- line 357:  Table 2 formatted and moved to a separate sheet to conform to journal requirements.  More information provided on the caption

Page 14 line 362:  Table 3 formatted and moved to a separate sheet to conform to journal requirements.  More information provided on the caption

Page 1 line 384:  Table 4 formatted and moved to a separate sheet to conform to journal requirements.  More information provided on the caption

Page 16 lines  394-395:  References provided

Page 17 line 413:  Table 5 formatted and moved to a separate sheet to conform to journal requirements.  More information provided on the caption, all elements on table formatted

Page 18 lines 422 and 433: Information of nature of field studies provided on line 434

Page 18  line  426-427:  References provided

Page 19 line 459: Table 6 formatted and moved to a separate sheet to conform to journal requirements.

Page 19 line 474:  More information provided to enhance discussion

All figure captions edited to conform to journal requirements.

Figure2 :  more information provided in the caption

Figure3:  Figure formatted properly and more information provided

Figure 4:  More information provided in the caption, Figure formatted properly

Figure 5:  More information provided in the caption, Figure formatted properly

Figure 9:  More information provided in the caption, coordinates added; Figure formatted properly

Figure10:  more information provided in the caption, Figure formatted properly, color legend provided